# Progressive cardiomyopathy with intercalated disc disorganization in a rat model of Becker dystrophy

Valentina Taglietti [ID][1]✉, Kaouthar Kefi[1], Busra Mirciloglu[1,2], Sultan Bastu[1], Jean-Daniel Masson [ID][1,2], Iwona Bronisz-Budzyńska[1,2], Vassiliki Gouni[3], Carlotta Ferri[3], Alan Jorge[1,2], Christel Gentil[4], France Pietri-Rouxel [ID][4], Edoardo Malfatti[1,5], Peggy Lafuste[1], Laurent Tiret [ID][1,2,7] & Frederic Relaix [ID][1,2,6,7]✉

## Abstract

**Becker muscular dystrophy (BMD) is an X-linked disorder due to in-frame mutations in the *DMD* gene, leading to a less abundant and truncated dystrophin. BMD is less common and severe than Duchenne muscular dystrophy (DMD) as well as less investigated. To accelerate the search for innovative treatments, we developed a rat model of BMD by deleting the exons 45–47 of the *Dmd* gene. Here, we report a functional and histopathological evaluation of these rats during their first year of life, compared to DMD and control littermates. BMD rats exhibit moderate damage to locomotor and diaphragmatic muscles but suffer from a progressive cardiomyopathy. Single nuclei RNA-seq analysis of cardiac samples revealed shared transcriptomic abnormalities in BMD and DMD rats and highlighted an altered end-addressing of TMEM65 and Connexin-43 at the intercalated disc, along with electrocardiographic abnormalities. Our study documents the natural history of a translational preclinical model of BMD and reports a cellular mechanism for the cardiac dysfunction in BMD and DMD offering opportunities to further investigate the organization role of dystrophin in intercellular communication.**

**Keywords** Becker Muscular Dystrophy; Connexins; Tmem65; Dilated Cardiomyopathy; Heart Failure
**Subject Categories** Methods & Resources; Molecular Biology of Disease; Musculoskeletal System

## Introduction

Muscular dystrophies are a group of inherited muscle conditions characterized by damage, weakening and detrimental remodeling of striated muscles over time, resulting in physical disability and cardiorespiratory complications. Among the different types of muscular dystrophies, Duchenne muscular dystrophy (DMD) and Becker muscular dystrophy (BMD) are two related diseases, both caused by mutations in the *DMD* gene encoding for dystrophin (Prior and Bridgeman, 2005). Dystrophin is part of the dystrophin-associated glycoprotein complex, which connects the internal cytoskeleton of muscle fibers to the extracellular matrix. This complex protects muscle fibers from the mechanical stress resulting from muscle contraction. Without dystrophin, myofibers become more susceptible to damage, leading to a progressive degeneration of muscle tissue and loss of muscle function (Duan et al, 2021). As explained by the reading frame rule (Koenig et al, 1989; Monaco et al, 1988), DMD is mainly caused by out-of-frame mutations with the consequent absence of dystrophin protein expression. On the other side, BMD is caused by in-frame mutations in the *DMD* gene that induce expression at highly variable extents of a truncated, partially functional dystrophin protein. BMD is less severe and has a lower incidence than DMD (Prior and Bridgeman, 2005). Animal models of DMD have provided crucial information on pathological mechanisms underlying disease progression and have been instrumental in the preclinical validation of new therapeutic approaches (Gaina and Popa Gruianu, 2021). Of note, there are comparatively few animal models of Becker muscular dystrophy to date (Heier et al, 2023; Hilton et al, 2023; Teramoto et al, 2020). Scarcity of animal models likely represent a bottleneck in translation of innovative therapeutic strategies. Indeed, among the hundreds of ongoing clinical trials for muscle dystrophies, only a few specifically target BMD (clinicaltrials.gov). To facilitate the development of novel therapeutic approaches for this condition, there is a need to a more precise description of the BMD disease trajectory. Understanding the clinical course of BMD becomes even more critical in the era of exon skipping gene therapy for DMD. Indeed, directed-exon(s) splicing approaches aim to positively restore the *DMD* transcript reading phase in every *DMD*-expressing cell, but doing so, they convert DMD patients into BMD patients. Because reading frame restoration may result in the loss of specific dystrophin domains, splicing choices should ideally be guided by evidence from genetic animal models for optimized benefit/risk ratios.

In this highly dynamic biomedical context, we have generated an original genome-edited rat model carrying a deletion of exons 45–47 (R-BMDdel45-47, hereafter named BMD rats), which is the most frequent deletion in BMD patients (Kaspar et al, 2009;

[1]Univ Paris-Est Créteil, INSERM, U955 IMRB, F-94010 Créteil, France. [2]École nationale vétérinaire d'Alfort, U955 IMRB, F-94700 Maisons-Alfort, France. [3]ADVETIA, Centre Hospitalier Vétérinaire, F-78140 Vélizy-Villacoublay, France. [4]Sorbonne Université, INSERM, UMRS974, Center for Research in Myology, F-75013 Paris, France. [5]APHP, Filnemus, EuroNMD, Centre de Référence de Pathologie Neuromusculaire Nord-Est-Ile-de-France, Departement Pathologie, Henri Mondor Hospital, F-94010 Créteil, France. [6]EFS, IMRB, F-94010 Créteil, France. [7]These authors contributed equally: Laurent Tiret, Frederic Relaix. ✉E-mail: valentina.taglietti@inserm.fr; frederic.relaix@inserm.fr

Takeshima et al, 2010; Vengalil et al, 2017). While there is already a BMD mouse model (Heier et al, 2023), previous analyses of DMD rat models have demonstrated that they provide a disease progression and severity much closer to the human condition compared to mouse models. This includes early-onset cardiomyopathy and precocious lethality. To characterize BMD rats on a disease severity scale, we have systematically compared BMD rats to their wild-type littermates and to the previously described R-DMDdel52 (hereafter named DMD rats; (Taglietti et al, 2022; Taglietti et al, 2023)). Here, we report that at one year of age, abnormalities in locomotor and diaphragmatic muscles observed in BMD are much milder than those in DMD rats. Most notably, BMD rats developed a progressive cardiomyopathy with reduced ejection fraction and pronounced fibrosis, closely mimicking the major cardiac involvement diagnosed in BMD patients carrying a 45–47 deletion. Mechanistically, we report a delocalization of TMEM65 and Connexin-43 associated with an ultrastructural disorganization of intercalated discs (ICDs) in BMD and DMD ventricles. TMEM65 and Connexin-43 are typically expressed at the ICDs and are necessary for establishing a fully functional electrical syncytium and adaptive sinus rhythm (Sharma et al, 2015; Teng et al, 2022). Altogether, we provide evidence that the functional support provided by the del45-47 truncated dystrophin in skeletal muscles of BMD rats is not transposable to the cardiac muscle, confirming distinct mechanisms of action between the two tissues and providing an accurate model for BMD preclinical evaluations.

## Results

### Generation of a Becker muscular dystrophy rat model

We generated a Becker muscular dystrophy rat model by inducing an in-frame deletion of exons 45–47 of the *Dmd* gene (Fig. 1A). Compared to their wild-type (WT) healthy littermates, male BMD rats appeared similar in size and showed no significant reduction in their weight and body mass index (BMI) (Table 1). BMD limb muscles were similar to those of WT, while those of DMD rats were severely atrophied (Table 1), as previously demonstrated (Taglietti et al, 2022). Interestingly, at both 6 and 12 months of age, the weight of BMD hearts was significantly higher than that of WT, while no significant difference was observed between WT and DMD hearts (Table 1). Next, we verified dystrophin protein levels by Western blot using the DYS1 antibody that specifically targets the ROD domain (encoded by exons 26–30) of the Dystrophin protein (Fig. EV1A). In both *tibialis anterior* (TA) and heart samples, we quantified a reduced Dystrophin abundance in BMD, while we confirmed its complete absence in DMD samples (Fig. 1B–D). In line with these results, we observed decreased Dystrophin immunofluorescence intensity at the sarcolemma on cross-sections of TA from BMD rats at 6 months (Fig. 1E). Dystrophin intensity, quantified as a percentage normalized to the WT controls, was markedly reduced in TA, diaphragm and heart of BMD rats, reaching an average of 33% of levels quantified in controls (Fig. 1F). This reduction in muscles was confirmed at 11 months using the antibodies DYS1 and DYS2 that specifically targets the C-terminal domain coded by exons 77–79 (Fig. EV1A–C). In DMD, we also confirmed the presence of rare spontaneously revertant myofibers (Fig. EV1D,E). Next, we

quantified the abundance of the two dystrophin-associated β-dystroglycan and nNOS proteins on TA sections. We found that β-dystroglycan was significantly reduced in BMD samples in comparison to controls, while it was undetectable in the DMD muscle (Fig. 1G,H). Both BMD and DMD TA exhibited lower sarcolemmal nNOS expression, accompanied by cytoplasmic enrichment of the signal (Fig. 1G). Global reduction in nNOS abundance was confirmed by Western blot in both BMD and DMD samples (Fig. 1I,J). Of note, *Nos1* (encoding nNOS) transcript abundance was also reduced in both BMD and DMD muscles (Fig. 1K), which correlated with an upregulation of the two *mir34c* and *mir708* microRNA (Fig. 1L) that are known to target NOS1-3' UTR sequence (Guilbaud et al, 2018).

### Histological and functional evaluation of limb muscles in BMD and DMD rats

Histological evaluation of TA muscle on sections stained with Hematoxylin & Eosin (H&E) and Sirius red revealed no major pathognomonic sign of muscular dystrophy in BMD rats. At both 6 and 12 months of age, there was no evidence of fiber size variability in BMD samples (Fig. 2A,B), while the DMD TA presented a significantly higher proportion of smaller fibers with a decreased cross-sectional area (CSA) at 6 months (Fig. 2B,C). In addition, while muscles of DMD rats contained extensive Sirius red-positive fibrotic infiltrates compared to WT controls, BMD TA showed no significant fibrotic deposition (Fig. 2D). To detect and quantify putative myofiber necrosis, we examined IgG uptake by damaged myofibers. Interestingly, both BMD and DMD muscles showed a similar number of IgG-positive myofibers at 6 and 12 months of age, with a significant increase compared to WT (Fig. 2E,F). We next evaluated the regenerative capacity of muscles by detecting the embryonic myosin heavy chain (eMHC) protein on TA sections from WT, BMD, and DMD rats. As previously shown, muscles of DMD rats contained higher levels of eMHC when compared to WT rats at 6 months, with a decreasing number of eMHC-positive myofibers upon disease progression (Taglietti et al, 2023) while no eMHC-positive fibers were found in BMD muscles (Fig. 2G,H). To assess muscle function and strength, we performed a forelimb grip test. We confirmed that compared to their healthy littermates, DMD rats performed significantly worse at both 6 and 12 months of age (Fig. 2I). Of note, BMD rats displayed a significant decrease in the maximal developed force when compared to WT rats only at 12 months (Fig. 2I), supporting a slowly progressive skeletal muscle weakness. To quantify muscular fatigue in rats in response to exercise, we also calculated a force maintenance index (FMI) after repetitive grip measurements at intervals of a few minutes. In both BMD and DMD rats, we quantified an early, significant decline in their capacity to maintain a maximal force during multiple repetitions (Fig. 2J). In conclusion, BMDdel45-47 rats appeared moderately weaker than their WT littermates. Myonecrosis was observed as well as intact myofiber repair capacities, yielding a rather normal muscle morphology at the histological level with no sign of fibrosis.

### Assessment of MuSC properties in BMD and DMD rats

To investigate a potential reduction in the number of muscle stem cells (MuSCs) in the two BMD and DMD models, the number of

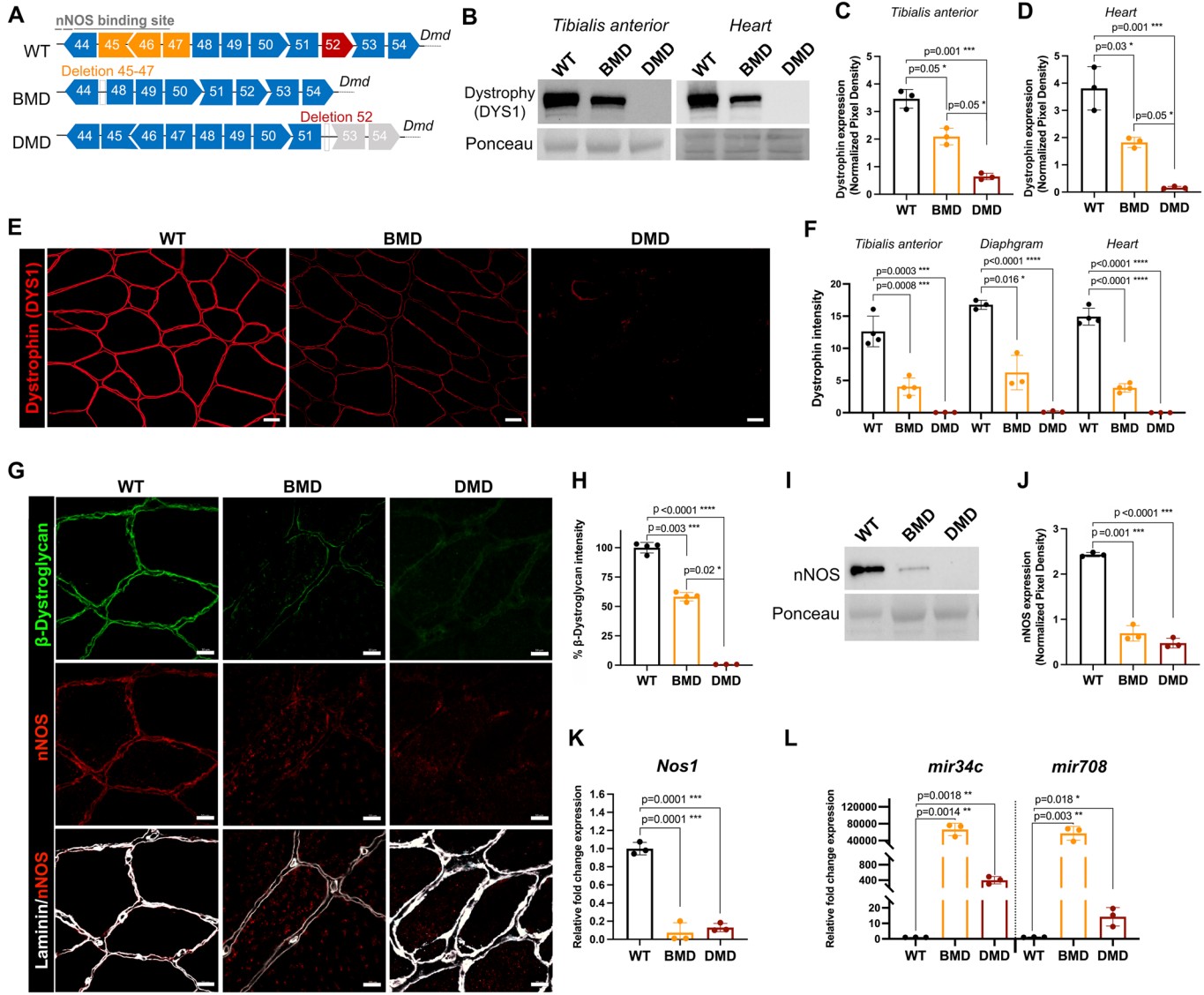

**Figure 1. Reduction and disorganization of the dystrophin-associated protein complex in muscles of *Dmd*-edited rat models.**

(A) Schematic diagram of a portion of the *Dmd* gene showing specific deletion of exons 45–47 and exon 52. (B) Western blot analysis for dystrophin from TA and heart of 6-month-old rats using the DYS1 antibody. (C, D) Quantification of Western blot analysis on TA (C) and heart (D). One-way ANOVA, $n = 3$. (E) Representative immunostaining of dystrophin in TA at 6 months of age. Scale bar 20 µm. (F) Normalized fluorescent signal intensity of dystrophin at 6 months of age. One-way ANOVA, $n = 4$ for TA and $n = 3$ for diaphragm and heart. (G) Immunostaining of β-dystroglycan (green), nNOS (red), and nNOS merge with laminin (white) in TA sampled from 6-month-old rats. Scale bar 10 µm. (H) Normalized fluorescent signal intensity of β-dystroglycan from (G). One-way ANOVA, $n = 4$. (I) Western blot analysis for nNOS in TA muscle extracts at 6 months. (J) Quantification of (I). One-way ANOVA, $n = 3$. (K, L) qRT-PCR quantification of *Nos1* (K), *mir34c* and *mir708* (L) expression. One-way ANOVA, $n = 3$. In (C, D, F, H, J–L), data are represented as mean ± SD. Source data are available online for this figure.

PAX7-positive cells was quantified (Heslop et al, 2000; Luz et al, 2002) and compared to that of WT littermates. Whatever the genotype of rats and independently of disease severity, statistical analysis revealed similar number of MuSCs (Fig. 3A,B). The double staining for PAX7 and Ki-67, a marker of proliferating cells, was used to reveal and quantify activated MuSCs. In the TA of DMD compared with that of WT, we found a higher number of PAX7 +:Ki-67+ MuSCs, while no activated MuSCs were found in BMD samples (Fig. 3C). We further investigated the proliferative capacity of MuSCs-derived myoblasts (PAX7 +:MYOD+) in vitro. No significant difference was found between WT and BMD,

showing that BMD myoblasts retain a normal proliferative potential. Conversely, DMD myoblasts showed impaired proliferative capacity, with a significantly lower number of PAX7 +: MYOD+ myoblasts co-expressing Ki-67 (Fig. 3D,E). To fully assess the myogenic program of the MuSC-derived lineage, we evaluated the expression of Myogenin (MYOG) on myoblasts cultured in a differentiation medium. After 5 days, we found no significant difference between WT and BMD, while a low number of MYOG-positive myocytes or myotubes was found in DMD samples (Fig. 3F). Also, BMD and WT myoblasts differentiated to the same extent (Fig. 3G,H), contrarily to DMD myocytes that formed

**Table 1. Morphometric and muscle weight data.**

| Parameter | 6 months | | | 12 months | | |
|---|---|---|---|---|---|---|
| | WT | BMD | DMD | WT | BMD | DMD |
| Body weight (g) | 628.3 ± 40.4 | 613.15 ± 17.5 | 518.7 ± 27.8* | 796.7 ± 52.4 | 746.7 ± 52.4 | 531 ± 61.2*** |
| Tibia length (mm) | 50.41 ± 0.73 | 50.95 ± 0.32 | 49.50 ± 1.23 | 50.41 ± 0.09 | 50.34 ± 0.04 | 50.30 ± 0.29 |
| Body mass index (g/cm²) | 0.78 ± 0.03 | 0.79 ± 0.04 | 0.62 ± 0.03*** | 0.98 ± 0.07 | 0.94 ± 0.04 | 0.67 ± 0.7** |
| Body length (cm) | 28.3 ± 0.4 | 27.8 ± 1.1 | 28.7 ± 1.1 | 28.5 ± 0.3 | 28.1 ± 0.2 | 28.1 ± 0.2 |
| Tibialis anterior corrected (g) | 1.03 ± 0.08 | 0.96 ± 0.07 | 0.88 ± 0.09$ | 1.25 ± 0.18 | 1.15 ± 0.04 | 0.80 ± 0.15** |
| Heart corrected (g) | 1.98 ± 0.11 | 2.21 ± 0.13* | 1.93 ± 0.15 | 1.77 ± 0.03 | 1.99 ± 0.09** | 1.83 ± 0.09 |
| EDL corrected (g) | 0.28 ± 0.07 | 0.26 ± 0.01 | 0.25 ± 0.05 | 0.31 ± 0.08 | 0.33 ± 0.05 | 0.16 ± 0.04* |
| Soleus corrected (g) | 0.28 ± 0.03 | 0.28 ± 0.04 | 0.23 ± 0.6* | 0.32 ± 0.04 | 0.32 ± 0.04 | 0.22 ± 0.03* |

Results are expressed as mean ± SD; $n \geq 4$.
P values refer to WT and have been calculated by a Tukey (*$p < 0.05$, **$p < 0.01$, ***$p < 0.001$) or Kruskal-Wallis ($ values) post-hoc test.

smaller myotubes (Fig. 3G), with a significant decreased fusion index as previously shown (Taglietti et al, 2023) (Fig. 3H).

## Histological and functional evaluation of the diaphragm in BMD and DMD rats

H&E and Sirius red staining showed an intermediate muscle degeneration and fibrosis in BMD samples compared to WT and DMD (Fig. 4A). There was a tendency towards fibrotic deposition in BMD compared to WT littermates at 6 months of age, that became significantly increased at 12 months (Fig. 4B). At 6 months, immunofluorescence analysis for eMHC showed a significant increased number of newly regenerated myofibers in BMD diaphragm, although at a significantly lower extent than in DMD samples that were prone to an intense regeneration in a more severe dystrophic context (Fig. 4C,D). To evaluate a potential ventilatory deficit in BMD rats, we performed whole-body plethysmography and identified that at 12 months of age, ventilatory parameters were not physiologically modified, contrarily to those of DMD rats that showed a significantly lower peak expiratory flow (PEF), peak inspiratory flow (PIF), and minute volume (MV) (Fig. 4E–J).

## Histological and functional evaluation of the heart in BMD and DMD rats

In humans, the majority of Becker patients develop heart failure at around 30 years, accompanied by electrocardiographic (ECG) defects, cardiomyocyte hypertrophy and interstitial fibrosis (Del Rio-Pertuz et al, 2022; Piccolo et al, 1994). We therefore paid particular attention to this part of the disease trajectory in BMD rats, extending our comparison with DMD rats. We first examined the heart of dystrophic and WT rats at the histological level by H&E and Sirius red staining. From 6 months onwards, interstitial fibrosis was elevated in DMD rats compared to WT (Fig. 5A,B), confirming our previous data (Taglietti et al, 2022) and those reported in patients (Shih et al, 2020). In 12-month-old BMD rats, interstitial fibrosis paralleled that of DMD samples, but it was less pronounced at 6 months (Fig. 5A,B). This suggested that the mechanism of fibrosis had a later onset in BMD compared to DMD, but that once initiated, it was highly progressive. Quantification of cardiomyocyte surface area revealed a significant and equivalent increase in BMD and DMD rats compared to WT, present at the age of 6 months and maintained at 12 months (Fig. 5C,D). To further evaluate whether the observed deleterious histological signs impacted the cardiac function, we first recorded the electrocardiograms (ECG) of WT, BMD and DMD rats at basal conditions. To maintain a physiological heart rate with no interference of an acute stress response, we performed ECG under an optimized light sedation and anesthesia of animals, yielding similar mean heart rate values between the three genotypes (WT, 367 ± 55 bpm; BMD, 359 ± 41 bpm; DMD, 366 ± 21 bpm; $p = 0.935$). Representative ECG recordings under basal conditions are shown in the upper panel of Fig. 5E. The main anomaly observed was the systematic presence of a notched T wave in DMD, probably reflecting an activation/repolarization delay in the ventricles, often associated with increased QT interval (Maruyama et al, 1995). Accordingly, the QTpc of DMD rats was significantly higher than that of WT, while the QTpc value in BMD compared with WT was unchanged in these basal conditions (Fig. 5F). To

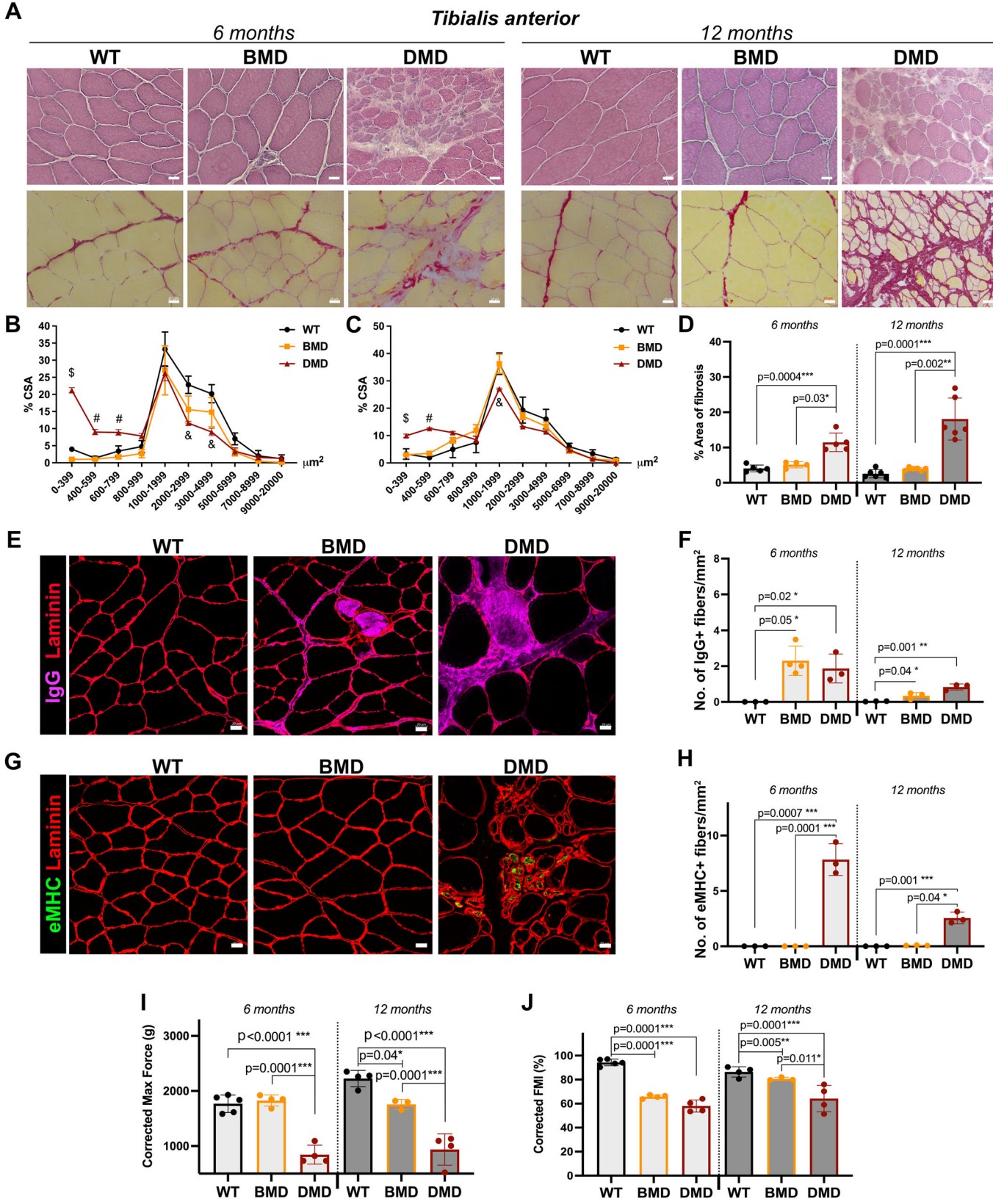

◀   **Figure 2.   Histological and functional defects in skeletal muscle of BMD and DMD rats.**

(A) H&E (top panel) and Sirius red (bottom panel) staining of TA cross-sections. Scale bar 20 μm. (B) Cross-sectional area (CSA) of TA from WT, BMD and DMD rats at 6 months of age. DMD were significantly different from WT: $, $p \leq 0.0001$; #, $p = 0.02$; &, $p = 0.007$. Two-way ANOVA, $n = 3$. (C) CSA of TA from WT, BMD, and DMD rats at 12 months of age. DMD were significantly different from WT: $, $p = 0.05$; #, $p = 0.001$ and &, $p = 0.03$. Two-way ANOVA, $n = 3$. (D) Fibrosis quantification referred to (A). One-way ANOVA, $n = 5$ for 6 months and $n = 6$ for 12 months. (E) Representative immunostaining of anti-rat IgG (purple) and laminin (red) at 6 months. Scale bar 20 μm. (F) Quantification of the number of IgG-positive myofibers. One-way ANOVA, $n = 3$. (G) Representative immunostaining of eMHC (green) and laminin (red) at 6 months. Scale bar 20 μm. (H) Quantification of the number of eMHC-positive myofibers. One-way ANOVA, $n = 3$. (I) Corrected maximal force by grip test. One-way ANOVA, $n = 4$–5. (J) Corrected force maintenance index (FMI) expressed as the percentage of force maintenance after the first grip measurement. One-way ANOVA, $n = 4$–5. In (B–D, F, H–J), data are represented as mean ± SD. Source data are available online for this figure.

challenge the heart, we further simulated a moderate sympathetic discharge by injecting the rats with a small amount of isoproterenol (ISO), a specific beta-adrenoreceptor agonist. In all treated rats and regardless of their genotype, isoproterenol produced a positive chronotropic effect with a ~20% increase in heart rate (Fig. 5F). No remarkable impact was observed on the overall ECG pattern of WT and DMD rats. On the contrary, we observed a modification of the T wave shape in BMD rats, and calculation of the QTpc indicated that it was significantly increased by isoproterenol (Fig. 5G). Altogether, these data showed that the dystrophin-deficient heart is susceptible to ventricular activation/repolarization abnormalities, which is already visible at rest in DMD rats, but only during a sympathetic-like solicitation in BMD rats.

## snRNAseq and Tmem65/connexin-43 evaluation on BMD and DMD heart

To reveal the cellular and transcriptional changes in cardiac samples, we performed snRNAseq on nuclei extracted from the heart apex from WT, BMD and DMD rats at 12 months of age. Apexes were dissected from fresh tissues and included a portion of septum, right and left ventricles (Dos Santos et al, 2020). Nuclei were FACS-sorted and processed for snRNAseq using 10X Genomics 5' Single cell technology. Unsupervised clustering revealed 11 cell populations (Fig. 6A), whose identity was validated by expression of cell-specific markers (Fig. EV2A,B). We found rather few alterations in the relative distribution of cell types when examining the densities of nuclei in the uniform manifold approximation and projection (UMAP) space (Fig. EV2C). More precise quantifications revealed that the relative distribution of cardiomyocytes was reduced in DMD samples, while macrophages were highly abundant in both BMD and DMD hearts. This was confirmed by immunostaining for the pan-macrophage marker CD68 (Fig. EV2D,E). When comparing diseased states (both BMD and DMD) to the healthy WT condition, different genes were deregulated specifically in cardiomyocytes. In line with the pathological phenotype, the upregulated genes were enriched for terms associated with dilated cardiomyopathy, cell and focal adhesion, actin binding fiber and heart development, while gene ontology terms as protein binding, transmembrane and acetylation were enriched in the list of downregulated genes (Fig. 6B). Notably, *Nppa* and *Nppb*, two markers of cardiac overload, were identified as the top genes with great upregulation in both BMD and DMD, while *Tmem65* and *Myh6* were strongly downregulated (Figs. 6C and EV2F). No changes in canonical markers of cardiomyocytes as *Ryr2* and *Gja1* (Gap junction alpha-1 protein coding for connexin-43) were observed by snRNAseq or qPCR (Figs. 6C and EV2F). Since the mRNA abundance of *Tmem65* was reduced in both BMD and DMD samples (Figs. 6C and EV2F) and given its role in the

intercalated disc (ICD) integrity (Teng et al, 2022), we sought to characterize TMEM65 expression and localization. Accordingly, we confirmed that TMEM65 protein abundance was significantly decreased in DMD samples compared to WT, while we found unchanged abundance in BMD heart (Fig. 6D–F). At the histological level, TMEM65 notably displayed a larger distribution and mislocalization from the ICD in both BMD and DMD ventricles (Fig. 6F). Since TMEM65 interacts and regulates Connexin-43 (Cx43) function in myocardium (Sharma et al, 2015), we found that Cx43 was more internalized in both BMD and DMD, with a lower co-localization with N-cadherin (NCAD), an intercalated disc marker (Fig. 6G–I). In addition, NCAD had a more imprecise localization in BMD and DMD samples compared to its straight and marked localization in WT (Fig. 6G–I). To deeper evaluate the ultrastructure of ICDs, transmission electronic microscopy (TEM) analysis was performed on left ventricles from WT, BMD and DMD rats. Both BMD and DMD ventricles showed fragmented and misaligned sarcomeres, with abnormal ICDs (Fig. 6J). Indeed, ICDs from BMD and DMD rats were characterized by a disorganization of the regions at the *adherens* junctions, often displaying a widened amplitude. Measurement of the intercalated disc amplitude (i.e., as indicated by the arrows in Fig. 6J) revealed a significantly higher amplitude in both BMD and DMD samples (Fig. 6K). A detailed examination of regions containing high-amplitude ICDs revealed convoluted regions with a notable detachment from the thin actin filaments typically seen within the folds of control ICDs (i.e., as indicated by the asterisks in Fig. 6J). Another common observation was the presence of structural irregularities of the interdigitations often accompanied by the accumulation of vesicles. This was consistent with TMEM65, NCAD, and Cx43 altered localization and underscored a crucial role of dystrophin in the ultrastructural organization of ICD, whose alterations likely contributed to the loss of electrical synchronization in cardiomyocytes revealed by ECG anomalies.

## Comparative morphological and contractile function assessment of the BMD heart

To evaluate the functional consequences of the aforementioned molecular and ultrastructural anomalies, we performed echocardiographic analyses of WT, BMD, and DMD rats. At 12 months, the M-mode image clearly showed significant ventricular dilatation in BMD rats, compared with WT (Fig. 7A,B), which was associated with a significant reduced left ventricular free wall thickness in diastole (LVFWd) (Fig. 7C). These features that are hallmarks of a dilated cardiomyopathy were not seen at 7 months (Fig. 7A–C). Similar aortic diameters (Ao) between genotypes confirmed the absence of variability in heart growth or stenosis, in addition to demonstrating the repeatability of quantified measurements,

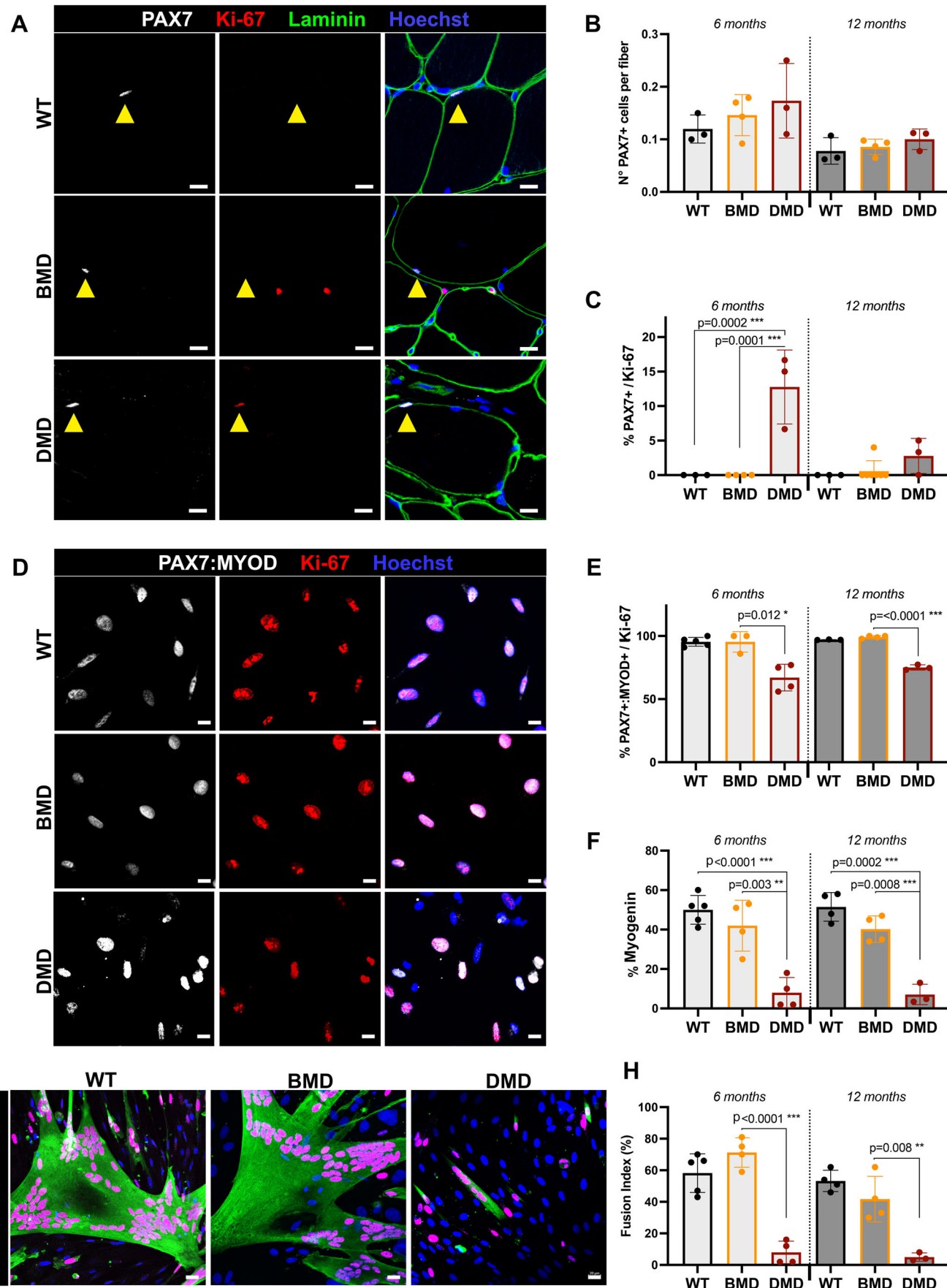

**Figure 3. Normal numbers of skeletal muscle stem cells with intact myogenic capacity in BMD rats.**

(A) Representative images of PAX7 (white), Ki-67 (red) and laminin (green). Scale bar 10 μm. (B) Quantification of the number of PAX7-positive MuSCs. One-way ANOVA, $n = 3$. (C) Percentage of activated (Ki-67+). One-way ANOVA, $n = 3$. (D) Immunostaining for PAX7 and MYOD (white), Ki-67 (red) and Hoechst (blue) on cultured myoblasts. Scale bar 10 μm. (E) Percentage of proliferative (Ki-67+) myoblasts. One-way ANOVA, $n = 3$–4. (F) Percentage of MYOG-positive cells. One-way ANOVA, $n = 4$–5. (G) Representative images of MYOG (purple), sarcomeric myosin heavy chain (MHC, green) and Hoechst (blue) on differentiated cultured cells. Scale bar 20 μm. (H) Fusion index referred to (G). One-way ANOVA, $n = 4$–5. In (B, C, E, F, H), data are represented as mean ± SD. Source data are available online for this figure.

whatever the age of the rats (Fig. 7D). To assess the systolic function of the left ventricle, we calculated the shortening fraction (LVFS) and ejection fraction (LVEF), revealing a notable reduction of both parameters in 12-month-old BMD rats (Fig. 7E,F). We observed heterogeneity in the group of BMD rats, suggesting an individualization of cardiac morbid trajectories, also observed in DMD rats (Fig. 7E,F). Five of the 6 BMD rats displayed low LVFWd values, contained within the first quartile of WT rat values, and for 2 of them the LVEF value indicated moderately to severely impaired systolic function (<50% LVEF; Fig. 7G). In line with low ejection fraction values and the high interindividual variability, we observed in some older BMD rats (>12 months) a sudden death or clinical signs of decompensation (dyspnea or orthopnea and cyanosis), prompting their compassionate killing. In both cases, necropsy revealed pulmonary edema and ascites that confirmed global congestive heart failure. Interestingly, we found an even more significant left ventricular systolic dysfunction in DMD rats at both 7 and 12 months (Fig. 7E,F), while LVIDd and LVFWd values remained within a range of subnormal values (Fig. 7B,C). This suggested that the slightly thinner myocardium of imaged DMD rats, fully deficient in dystrophin, was precociously weak. Notably, clinical heart failure was not reported in DMD rats, whose severe skeletal muscle damage progressing to cachexia resulted in compassionate killing likely anticipating the expression of congestive clinical signs. Altogether, these data show a convergence of cardiac disease patterns in BMD and DMD rats, characterized by a progressive systolic dysfunction. While the onset is earlier in DMD rats, the severity is equally evident in BMD rats at one year of age, developing a dilated cardiomyopathy progressing to heart failure with reduced ejection fraction, according to an individual temporal trajectory.

We compared those data to those of a cohort of five BMD patients carrying a deletion of exons 45–47 who are followed in our Reference Center. All five patients had contractile or morphological abnormalities on echocardiography, while ECG alterations were reported in three of them (Table 2). More precisely, we found ventricular hypokinesia, left ventricular dilatation, hypokinesia and reduced ejection fraction, averaging 29.3%. The mean age at onset of cardiac function abnormalities was 33 years. These findings emphasized that our BMD rat model strongly mimics the human orthologous disease trajectory.

## Discussion

Becker muscular dystrophy is less common than Duchenne muscular dystrophy. It manifests later, at an average age of 31.5 years, and in 40% of cases it begins with clinical signs suggestive of heart failure. Gait abnormalities are observed in 54% of patients at an average age of 37 (Nakamura et al, 2023). Deletion of exons 45 to 47 is the most frequent mutation in BMD patients, accounting

for roughly 23% of cases. Del45-47 patients never require ventilatory assistance, but they share a series of cardiac abnormalities. Around 20% of del45-47 patients will drop to an ejection fraction below 55% (Nakamura et al, 2023), prompting standard treatment to limit heart failure symptoms (Shih et al, 2020). For the most severe BMD cases, heart transplantation has demonstrated its effectiveness in the long-term survival of patients (Wells et al, 2020). However, preventive rather than symptomatic treatments are needed, whose development is hampered by the lack of precise knowledge of the underlying pathogenic mechanisms. Since human cardiac biopsies are rare, it is challenging to identify these mechanisms in patients. Animal models of BMD will be invaluable in elucidating these mechanisms, but such experimental models have only recently been introduced. To the best of our knowledge, previous reports described two of them: a rat model, designed here as BMDdel3-16, with an in-frame deletion spanning exons 3–16 (Teramoto et al, 2020), and a mouse model generated by deletion of the exons 45–47, termed *bmx* mice (Heier et al, 2023). Both models exhibit muscle histopathological alterations, including fibrosis. However, they differ in terms of functional readouts. *Bmx* mice showed hypertrophied skeletal muscles and reduced muscle functions by the age of 2.5 months, and they were diagnosed with heart failure characterized by a 28% reduction in ejection fraction at 18 months (Heier et al, 2023). On the contrary, BMDdel3-16 rats showed neither muscle nor cardiac dysfunction (Teramoto et al, 2020). In this study, we present a BMDdel45-47 rat model in the Sprague Dawley outbred background, generated by inducing an in-frame deletion of exons 45–47 of the *Dmd* gene. This model faithfully recapitulates key manifestations of the human disease, including cardiac impairments that precede the later-onset skeletal muscle alterations. Given that cardiomyopathy is the primary cause of death in BMD patients (Connuck et al, 2008; Steare et al, 1992; Yilmaz et al, 2008), and it anticipates the skeletal muscle decline (Finsterer and Stollberger, 2008; Mavrogeni et al, 2015), the study of the cardiac phenotype holds significant relevance. Our analysis of R-BMDdel45-47 hearts faithfully parallels the human cardiac disease trajectory, evidenced by cardiomyocyte hypertrophy and ultrastructural organization, myocardial fibrosis and contractile dysfunction leading to a progressive dilated cardiomyopathy with reduced ejection fraction. The failing heart with a low ejection fraction is a hallmark of BMD in human patients lacking at least exons 45–47 (Finsterer and Stollberger, 2008; Ho et al, 2016; Mavrogeni et al, 2015; Nigro et al, 1995; Steare et al, 1992; Suselbeck et al, 2005), and the review here of five additional unpublished BMD patients confirmed this international dataset. The resulting dilated cardiomyopathy (DCM) is prevalent in both BMD and DMD patients, eventually culminating in congestive heart failure or cardiac arrest (Florczyk-Soluch et al, 2021; Nigro et al, 1990; Wittlieb-Weber et al, 2020). The cardiac phenotype occurred between 7 and 12 months in BMD rats, corresponding in humans to the transition between young (30 years) to middle (50

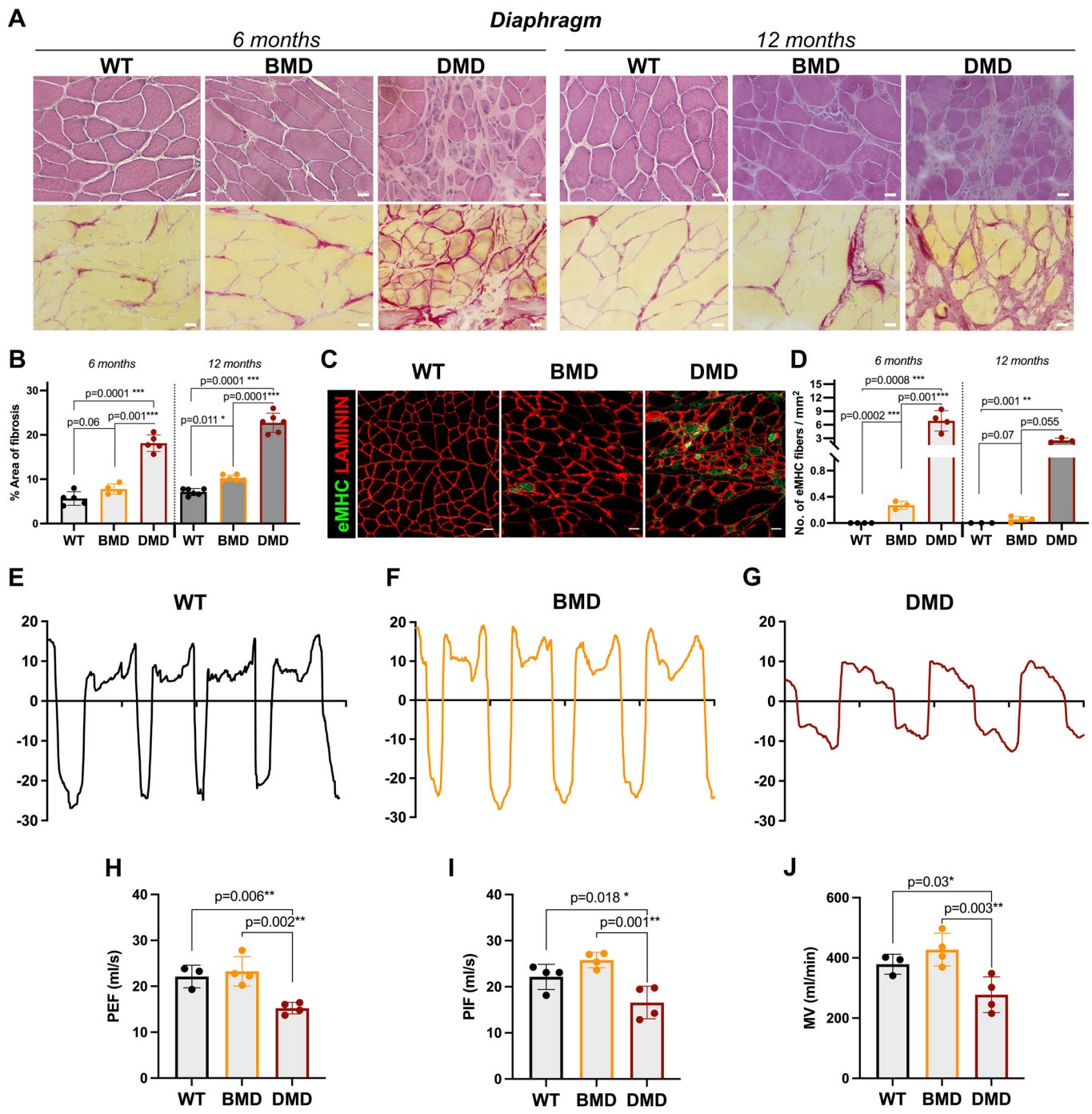

**Figure 4. Histological and functional evaluation of the diaphragm in BMD and DMD rats.**

(A) H&E (upper panel) and Sirius red (lower panel) staining. Scale bar 20 μm. (B) Fibrosis quantification. One-way ANOVA, $n = 5$–6. (C) Representative immunostaining of eMHC (green) and laminin (red) in at 6 months of age. Scale bar 20 μm. (D) Quantification of the number of eMHC-positive fibers. One-way ANOVA, $n = 4$. (E–G) Representative ventilatory flow profiles of 12-month-old WT (E), BMD (F), DMD (G). (H–J) Peak expiratory flow (PEF) in ml/s (H), Peak inspiratory flow (PIF) in ml/s (I), minute volume (MV) in ml/min (J). One-way ANOVA, $n = 4$. In (B, D, H–J), data are represented as mean ± SD. Source data are available online for this figure.

years) adulthood (Ghasemi et al, 2021), paralleling the reduced ejection fraction noted in del45-47 patients around the age of 50. The chronology of the trajectory is therefore finely respected in BMD rats. Surprisingly, myocardial thickness in BMD rats is significantly reduced earlier than in DMD rats, and may be due to a

sustained skeletal muscles' strength of BMD rats, which exercise more than DMD between 7 and 12 months of age, prompting constant contraction stress in their heart.

DCM-specific cell states were also highlighted by snRNAseq and confirmed by qPCR, revealing BMD and DMD cardiomyocyte

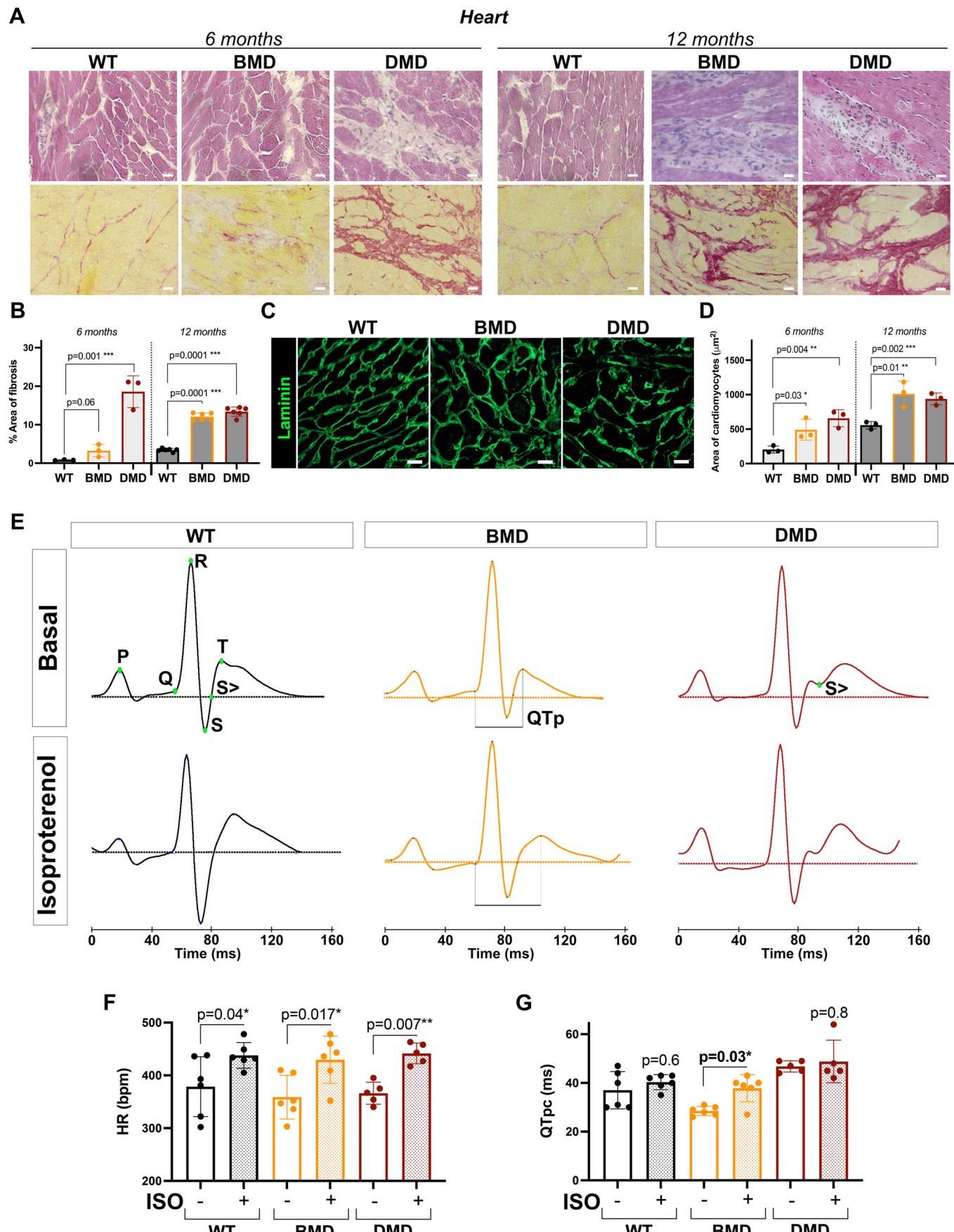

**Figure 5. Hypertrophic remodeling and electrical abnormalities in the heart of BMD and DMD rats.**

(A) H&E (upper panel) and Sirius red (lower panel) staining. Scale bar 20 μm. (B) Fibrosis quantification. One-way ANOVA, $n = 3$ for 6 months and $n = 6$ for 12 months. (C) Representative immunostaining for laminin. Scale bar 20 μm. (D) Quantification of cardiomyocytes size. One-way ANOVA, $n = 3$. (E) Representative ECG signals from WT, BMD and DMD at 7–8 months of age under basal conditions (upper panels) and after an isoproterenol challenge (lower panels). The isoelectric line and the P-QRS-T wave legend are shown. The QTp interval is indicated in BMD rat tracings. (F, G) Heart rate (HR) in bpm (F) and the QTpc in ms (G), before (−) and after (+) injection of isoproterenol (ISO). One-way ANOVA, $n = 6$. In (B, D, F, G), data are represented as mean ± SD. Source data are available online for this figure.

transcriptional changes. An increase in *Nppa* and *Nppb*-expressing cells and significant reduction in *Myh6* levels resembled those shown in human DCM ventricles (Koenig et al, 2022; Wang et al, 2020). At the ultrastructural level, BMD and DMD left ventricles showed alterations previously reported in DCM human hearts with deterioration of ICDs (Ito et al, 2021). The ICD comprises *adherens* junctions, desmosomes and gap junctions, serving as a crucial structure linking neighboring cardiomyocytes, and enabling mechanical and electrical coupling for contraction force transmission. Both BMD and DMD left ventricles exhibited disorganized ICDs with irregular interdigitations and wider amplitude that impact on nearby sarcomeres, as evidenced by actin detachment and sarcomere disruption in proximity to the *adherens* junctions. Early predisposition of BMD rats to ECG abnormalities, evidenced by a prolonged QTpc following isoproterenol exposure, was unrelated to DCM, but it is in line with ICD disorganization and may be directly linked to the left ventricular dysfunction and arrhythmia observed in BMD patients (Finsterer and Stollberger, 2008; Nigro et al, 1995; Steare et al, 1992). While the ECG irregularities in dystrophic hearts have been historically ascribed mainly to endomyocardial fibrosis, cardiomyocytes hypertrophy or disbalance of the autonomic nervous function (Maeda et al, 1995; Silva et al, 2007), our findings point to an additional intrinsic causative factor. We propose that disruption in the localization of TMEM65 and Cx43 could play a role in altering electrical synchronization during myocardial depolarization or repolarization, due to partly inefficient intercellular communications. Indeed, we demonstrated that TMEM65 was mislocalized in both BMD and DMD heart samples. TMEM65 has been reported to provide a physical platform for NaV1.5 and this interaction is necessary for the localization of both NaV1.5 and Cx43 to the ICDs with the eventual formation of Cx43 functional channels for electrical and chemical communication between cardiomyocytes (Teng et al, 2022). In our dystrophic samples, TMEM65 disorganization led to Cx43 internalization, impairing cardiac conduction and functions. We can also suggest that in BMD rats, the reduced amount of Dystrophin, the truncated produced isoform, or both may induce a membrane addressing defect of proteins dedicated to intercellular connection. Aside from the quantitative defect, this may highlight the functional role of the nNOS interaction domain, or the functional alteration of the 3D structure of the truncated dystrophin. Independently of the frequently cited role of Dystrophin in protecting against the mechanical stress imposed by sarcomere contraction, these data shed light on the ancestral role of dystrophin in molecular clustering and formation of adhesion complexes at the membrane (Mirouse, 2023).

While we did not confirm a reduced protein abundance corresponding to the decreased mRNA levels of *Tmem65* in both BMD and DMD heart tissues, the delocalization of TMEM65 at the intercalated disks could explain the dystrophic cardiac phenotype that

is reminiscent of the phenotype induced by knocking down *Tmem65* (Teng et al, 2022). Since the defect in complete localization of Cx43 has been associated with distinct forms of dilated heart caused by *Lamin A/C* or *Dmd* mutations (Gonzalez et al, 2018; Gonzalez et al, 2015; Le Dour et al, 2022), our findings underscore a shared mechanism contributing to DCM in those various genetic cardiomyopathies. Consequently, gaining insights into the regulatory dynamics of Cx43 and TMEM65 will be crucial for the identification of new therapeutic targets with broad relevance.

The natural history of the BMD rat disease revealed milder histological and functional impairments of skeletal muscles when compared to DMD. Functionally, BMD rats exhibited a reduced muscular force, particularly pronounced at later life stages, with a significant predisposition to fatigue. Regarding respiration, they presented with a respiratory function similar to healthy subjects, as reported in BMDdel45-47 patients (D'Angelo et al, 2011; Lo Mauro and Aliverti, 2016). The maintenance of skeletal muscle function in BMD rats contrasts with the deterioration observed in DMD rats, whose resting ventilatory flow was found here to be reduced at 12 months, in line with the major symptoms of DMD patients. Finally, BMD rats recapitulate the human disease in terms of reduced dystrophin abundance (Beggs et al, 1991; Hoffman et al, 1989; Kesari et al, 2008; van den Bergen et al, 2014). In BMDdel45-47 patients, dystrophin levels are variable and ranked between 5 and 76% (Fiorillo et al, 2015; Kesari et al, 2008; van den Bergen et al, 2014). In BMD rats, the reduced level of dystrophin is correlating with deregulation of β-dystroglycan and nNOS, highly resembling findings in BMD patients (Lai et al, 2009; Nicolas et al, 2015). Precise mechanisms underlying dystrophin reduction in BMDdel45-47 muscles remain to be fully understood, but may involve protein instability due to domain disorganization (Delalande et al, 2018; Nicolas et al, 2015) or dystrophin-targeting microRNAs levels, regulated by inflammation and glucocorticoids (Fiorillo et al, 2015; McCormack et al, 2023). On the other hand, while a putative enhanced proteasomal activity was excluded in vitro (Teramoto et al, 2020), the involvement of other protein degradation systems remains to be assessed. Thus, more studies are still warranted to elucidate these mechanisms. This data will be instrumental to optimize dystrophin restoration therapies in BMD patients, or in DMD patients converted to BMD after exon skipping.

In conclusion, this work describes a unique and highly mimetic rat model of all aspects of the human Becker muscular dystrophy, and reveals a common mechanism with DMD, namely a failure in the cardiomyocyte to address key intercellular communication proteins to the membrane. This model is relevant for a more precise analysis of key pathogenic steps of the natural history of BMD and the underlying molecular mechanisms. This BMD rat model will be invaluable for preclinical evaluation of innovative precision therapies targeting the heart.

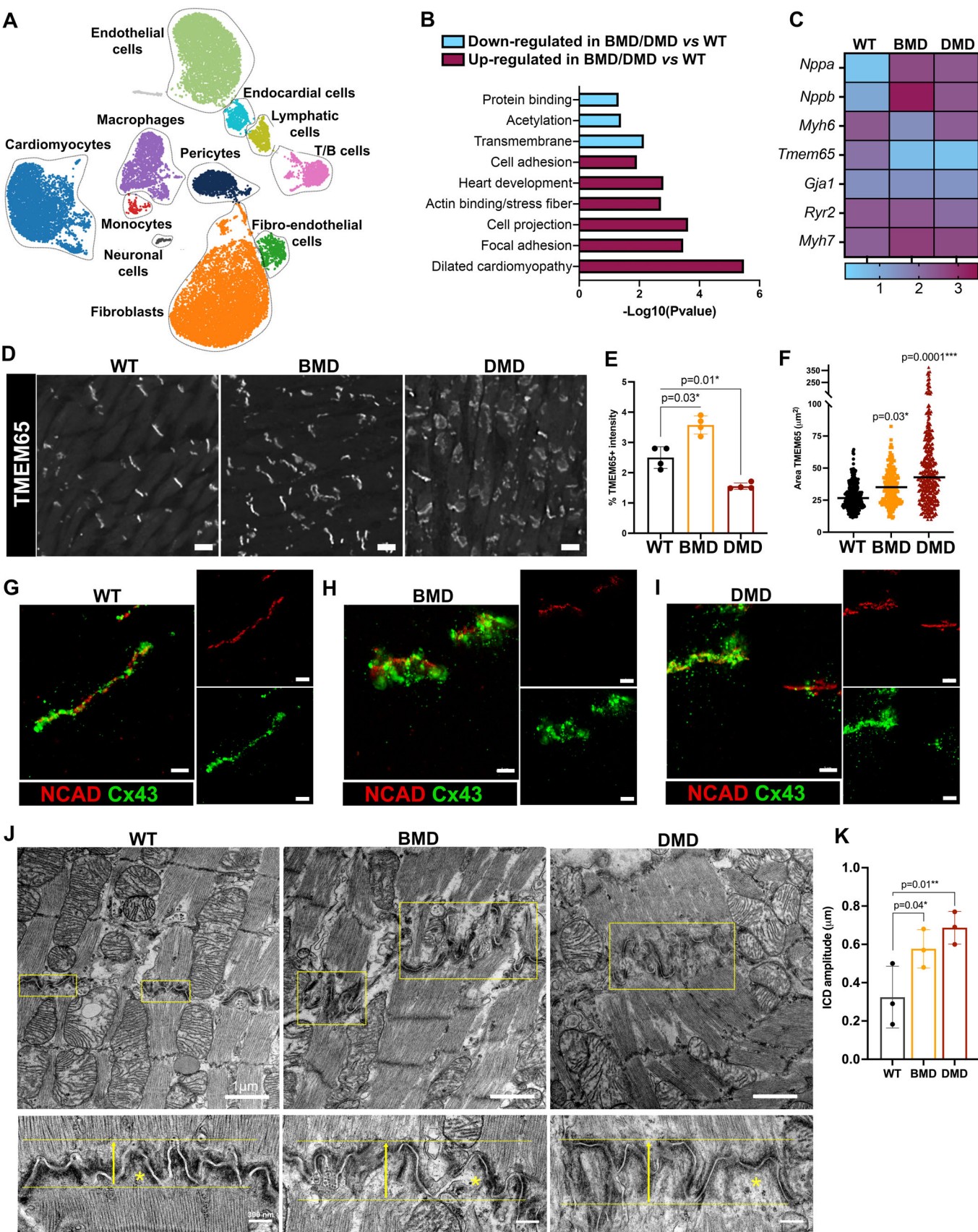

**Figure 6. Abnormal lateralization of TMEM65 and connexin-43 proteins at the ICDs in BMD and DMD rats.**

(A) UMAP of resident cell populations clusters, identified according to specific markers (Fig. EV2). (B) Gene ontology analysis of down- and up-regulated genes. (C) Heat map showing differentially expressed genes by snRNAseq. (D) TMEM65 immunostaining in left ventricles of 12-month-old rats. Scale bar 20 μm. (E, F) Quantification of TMEM65 intensity (E) and TMEM65-positive area (F) represented as the mean from biological triplicates. One-way ANOVA, $n = 3$–4. (G–I) NCAD (red) and Cx43 (green) immunofluorescence in WT (G), BMD (H), and DMD (I) left ventricles. Scale bar 5 μm. (J) TEM of the cardiomyocytes of WT, BMD and DMD left ventricles (scale bar 1 μm) at 6 months of age, with magnified images (scale bar 300 nm) of intercalated disks (ICDs), highlighted by yellow rectangles and lines. Asterisks show thick filaments in abnormal fold regions for BMD and DMD samples, compared to WT. (K) Quantifications of ICD amplitude average in μm of WT, BMD, and DMD left ventricles at 6 months of age. Quantified ICD amplitude is marked by arrows in (J). One-way ANOVA, $n = 3$. In (E, F, K), data are represented as mean ± SD. Source data are available online for this figure.

## Methods

### Study approval

All reported experiments were performed in accordance with the 2010/63/EU European Directive for the care and use of laboratory animals, transposed into French law (decree no. 2013-118). Our project was approved by the French Ministry of Higher Education, Research and Innovation (APAFIS#2022021916313160 and APA-FIS#25606-202005311746599) after constructive discussions and a positive review by our local animal welfare office (SBEA) and ethics committee (French ComEth no. 16).

### Generation and housing of R-BMDdel45-47 rats

Deletion of exons 45 to 47 of the rat *Dmd* gene by CRISPR/Cas9 was performed by using two pairs of sgRNAs on both sides of exons 45 and 47. Guide RNA selection was performed with CRISPOR (Concordet and Haeussler, 2018) on exon 45 (ENSRNOE00000395444) and exon 47 (ENSRNOE00000395442). The following sgRNAs were used: TTTGTAAGCTTGTCAGCTAGAGG (5'sgRNA, gR71), CTCAGTTC AGCTCAGGGACTGGG (5'sgRNA, gR72), ACAAAGCAATTGGTG ATGGTGGG (3'sgRNA, gR62), TAACACAAAGCAATTGGTGA TGG (3'sgRNA, gR70). The number attributed to each sgRNAs refers to the MIT specific score. Three μM of spCas9 and 8.8 μM of each sgRNAs were electroporated in rat Sprague Dawley (RjHan:SD provided by Janvier Labs, France) fertilized oocytes. Genotyping PCR and sequencing were used to confirm the deletion of exons 45–47 in F0 founder animals. For the genotyping, DNA was extracted and purified from ear biopsies and PCR reactions were prepared with Dream Taq buffer (ThermoFisher), 0.2 mM dNTP (ThermoFisher), 0.3 μM of each primer and 1.25 U Dream Taq DNA Polymerase (ThermoFisher). Three primers were used: CACCTTTGTATTGACAACCAAGTAA (F1), AATAGTAATTCTGCATCACATCACT (R6) and CCATCCT GGAGTTCCTGTAAGCCAC (R1). The primers F1 and R6 amplified the product of the deletion of the exons 45–47, while F1 and R1 amplified the WT allele. Thermal cycling conditions were as follows: initial denaturation at 95 °C for 3 min, 30 cycles with denaturation step at 95 °C for 30 s, annealing at 62 °C for 30 s, extension at 72 °C for 45 s, and the final extension at 72 °C for 5 min. Rats were housed in a pathogen-free, temperature-controlled environment (22 ± 2 °C) with a 12h–12 h dark/light photocycle.

### Grip test

Grip test of forelimbs was carried out using an isometric sensor (Bioseb). Four successive measurements were performed with a minimal 5-min rest period between each attempt. The force maintenance index (FMI), expressed as a percentage, was calculated by dividing the mean of the second, third and fourth measurements by the first measurement, in this cohort equivalent to the maximal force developed by each rat.

### Whole-body plethysmography

Ventilatory parameters were assessed by whole-body plethysmography (Emka Technologies) on conscious rats. Data were then recorded for at least 20 min and the analysis was performed using the iox2 software. Peak inspiration and expiration flow (PIF, PEF) represent, respectively, the maximum negative and positive flow during inspiration (downward trace) or expiration (upward trace). MV is the total volume breathed in a minute, calculated as the tidal volume multiplied by the ventilatory movement rate.

### ECG and isoproterenol challenge

Seven- to eight-month-old rats were sedated and lightly anesthetized with a subcutaneous injection of 5 mg/kg midazolam and 40 mg/kg ketamine 20 min before the ECG. Rats were maintained under very light anesthesia by mask inhalation of 0.5 to 1% isoflurane. On shaved and depilated skin, two electrodes were positioned, one on the right cervical dorsal region and the other ventral, where the cardiac apex strikes the ribs. Signals of this bipolar DII lead were acquired using a non-invasive telemetry system from Emka Technologies (rodentPACK) over a 10 min period (basal), at a sampling rate of 1000 Hz. Then, rats were injected subcutaneously with isoproterenol (Sigma I-5627) diluted in ultrapure water at 0.3 mg/kg, and 5 min later, at 1 mg/kg. Recordings were maintained until the heart rhythm came back to the basal condition.

ECG analysis was performed using ECGavg (v2) software and the "Averaged beats analysis" plugin that we developed with EMKA. The average ECG trace was obtained from 500 to 1500 cardiac cycles. The isoelectric line was set up as the voltage of the TP segment; the end of the S wave was defined as the intersection between the trace and the isoelectric line, or in the case of notched T-wave, as the point of return to the horizontal. QTpeak corrected (QTpc) values were calculated with the Bazett's derived formula $QTpc = QTp/(RR/f)1/2$, derived from a previous study (Kmecova and Klimas, 2010), with f equal to the mean RR interval of each evaluated rat. Here, we calculated f for each rat and condition.

### Echocardiography

Rats were sedated and lightly anesthetized with 5 mg/kg of midazolam and 50 mg/kg of ketamine. Three ECG electrodes were connected to the animal, and all examinations were performed by use of an ultrasonographic units equipped with a 12S-RS transducer

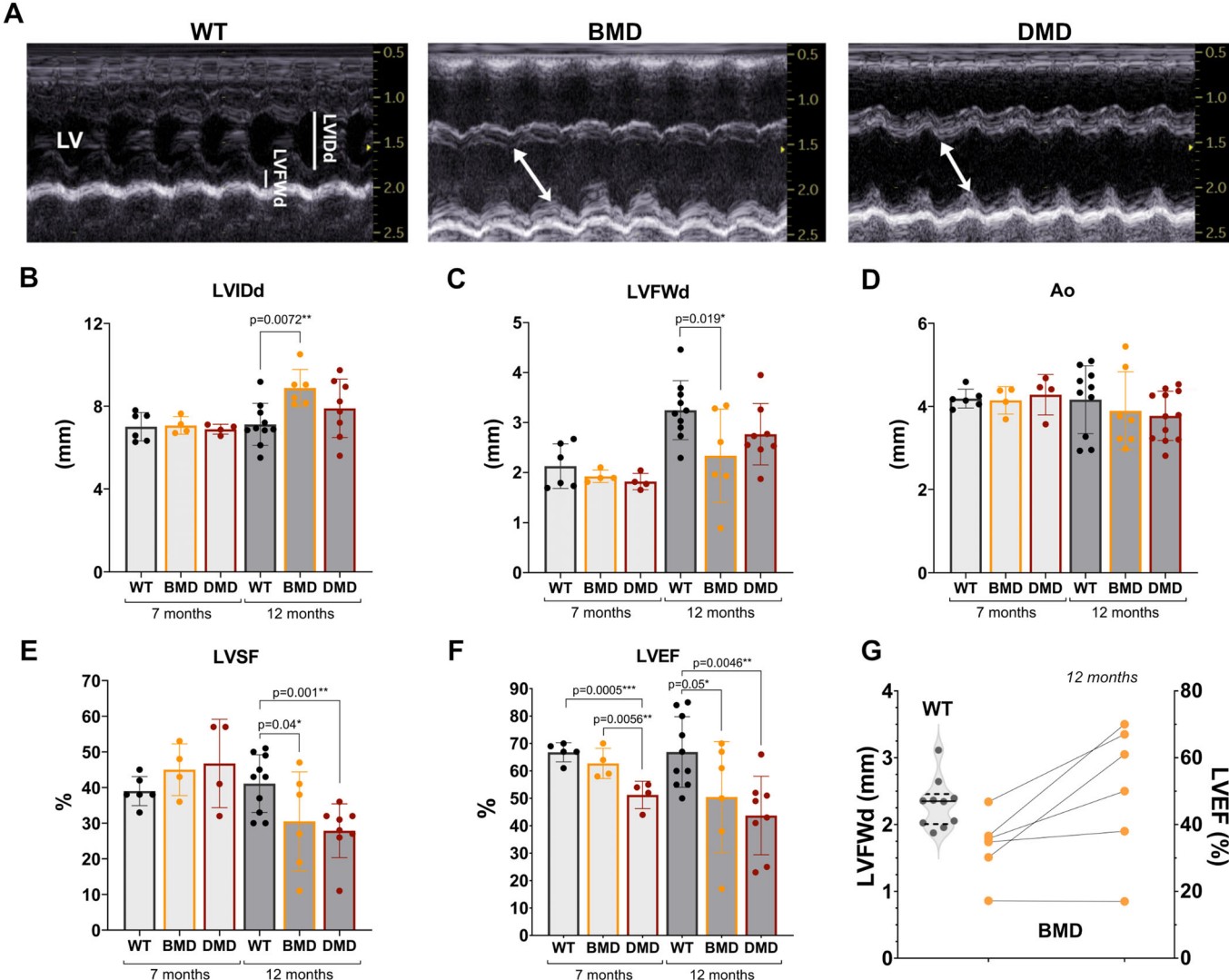

**Figure 7. Progressive development of a dilated cardiopathy with reduced ejection fraction in BMD and DMD rats.**

(A) Representative M-mode images of right and left ventricle (LV) at 12 months. Contractile asynchrony is indicated by an arrow. (B–F) Echocardiographic data for LV internal diameter at diastole (LVIDd, B), LV free wall thickness at diastole (LVFWd, C), aorta diameter (Ao, D), LV fractional shortening (LVFS, E) and LV ejection fraction (LVEF, F). One-way ANOVA, $n = 4$–6 at 7 months and $n = 6$–11 at 12 months. (G) Individual LVFWd and LVEF values for BMD rats whose LVFWd value is less than or equal to the last quartile of LVFWd values in WT rats (≤2.8 mm; the 2 lowest values in the WT group are indicated by gray symbols). An ejection fraction below 50% is considered mild dysfunction, progressing to severe dysfunction below 30%. One-way ANOVA, $n = 4$–6 at 7 months and $n = 6$–10 at 12 months. In (B–F), data are represented as mean ± SD. Source data are available online for this figure.

(Vivid™ iq, GE Medical System, Horten, Norway) or a MS-250 transducer operating at 21 MHz with 100 images/s (Vevo 2100-C, Fujifilm VisualSonics, Amsterdam, Netherlands). Analysis was performed using the EchoPac Software (GE Medical Systems) or the Cardio module from VisualSonics. M-mode and two-dimensional echocardiographic examinations were conducted. The right parasternal short-axis view was used to measure the right and left ventricular internal diameters at end-diastole, the left ventricular internal diameters at end-systole, as well as interventricular septum at end-diastole and end-systole, left ventricular free wall at end-diastole and end-systole, and left ventricular fractional shortening. The left ventricle ejection fraction was calculated by the Simpson's method of disc on a right parasternal four- or five-

chamber view and confirmed using the LV traces from the B-mode. The aortic diameter was measured from the right parasternal short-axis view. For each rat and each parameter, the value used for interindividual statistics was the mean of values measured on three different cycles. All analyzed rats maintained a minimal physiological heart rate value of 325 bpm or higher, without significant difference between the 3 genotypes ($p = 0.458$; one-way ANOVA).

## Human cardiac data

Medical records, including ECG and echocardiographic parameters, were reported from five BMDdel45-47 patients followed at the Neuromuscular Center of the Henri Mondor Hospital (France).

**Table 2.** Genetic and clinical data of BMD patients.

| Patient | Dystrophin mutation | Age of onset of cardiac dysfunction (years) | Cardio-respiratory symptoms | ECG alteration | Echocardiography |
|---------|---------------------|---------------------------------------------|------------------------------|----------------|-------------------|
| P 1 | del45-47 | 36 | Palpitations | Negative T waves | Posterior and basal hypokinesia |
| P 2 | del45-47 | 35 | No | No | LV dilatation, global hypokinesia. LVEF: 26% |
| P 3 | del45-47 | 25 | Syncopal episodes | Negative T waves | LV dilatation. LVEF: 31% |
| P 4 | del45-47 | 34 | Dyspnea | PR interval: 129 ms | Inferior hypokinesia |
| P 5 | del45-47 | 36 | Palpitations | Nr | LVEF: 31% |

*ECG* electrocardiography, *LV* left ventricle, *EF* ejection fraction, *Nr* not reported.

Data were acquired according to the American Society of Echocardiography guidelines (Cheitlin et al, 2003) in the routine care of patients, regularly monitored in a preventive cardiology unit.

## Histological staining

Muscles were immediately frozen in liquid nitrogen-cooled isopentane and sectioned at the cryostat to obtain 7-μm-thick sections. For H&E staining, nuclei were stained by immersing the slides in Mayer's haematoxylin (Sigma-Aldrich) for 3 min followed by rinsing in deionized water and dipping in lithium carbonate solution. Cytoplasm was stained by immersing slides in Eosin. After a wash in distilled water, slides were dehydrated in 50% and 70% EtOH, followed by equilibration in 95% and 100% EtOH. For Sirius Red staining, thawed cryosections were dipped in 90% EtOH for 2 min, stained in picro-sirius red solution (Sigma-Aldrich) for 25 min and then washed in water, followed by progressive dehydration in three changes of 100% EtOH.

H&E and Sirius Red dehydrated sections were transferred from 100% EtOH to 100% xylene and mounted with Eurokitt.

## Cell culture

Hindlimb muscles were harvested and digested in 3 U/ml Dispase II (Roche) and 0.5 U/ml Collagenase A (Roche) at 37 °C for 1 h. Lysate cells were centrifuges for 5 min at $80 \times g$ to eliminate debris and the suspension was filtered with 70 μm cell strainer before pelleting the cells by centrifugation. Obtained cells were cultured on Matrigel (Life Sciences) in 20% fetal bovine serum in DMEM supplemented with Glutamine and Penicillin/Streptomycin. After 48 h of culture, a set of cells was analyzed for the proliferation phase, while the other set was switched to 5% horse serum to induce the differentiation. Differentiated myotubes were analyzed after 5 days in differentiation medium.

## Cell culture immunostaining

Cells were fixed with 4% PFA at 4 °C for 10 min, and permeabilized with 0.5% Triton in PBS for 12 min at room temperature. The blocking was performed with 10% BSA for 45 min, before adding overnight primary antibodies: anti-MYOD (Dako, M3512), anti-PAX7 (Santa Cruz Biotechnology, sc-81648), anti-Ki-67 (Abcam, sp6 ab16667), anti-Myogenin (Santa Cruz Biotechnology, sc-52903) and anti-MHC chain (DSHB, MF20). After three washes, the cells

were then incubated with Alexa fluor secondary antibodies and Hoechst (Sigma-Aldrich, B2261) for 45 min and mounted. Pictures were acquired using a LSM800 confocal microscope (Zeiss).

## Immunofluorescence and quantifications

Muscle sections were defrosted at room temperature and permeabilization was performed with 0.5% Triton in PBS for 12 min, followed by blocking in 10% BSA in PBS for 45 min. Primary antibodies were diluted in 0.1% BSA in PBS and incubated overnight at 4 °C. The following primary antibodies were used: anti-dystrophin (Leica, NCL-DYS1 recognizing exons 26–30 of the ROD domain, and NCL-DYS2 targeting exons 77–79 of the C-terminal domain; Fig. EV1), anti-β-dystroglycan (Leica, B-DG-CE), anti-nNOS (Thermofisher, Catalog #61-7000), anti-eMHC (Santa Cruz Biotechnology, sc-53091), anti-laminin (Sigma-Aldrich, L9393), anti-Connexin-43 (Millipore, C6219), anti-N-cadherin (Sigma-Aldrich, C3865), anti-TMEM65 (EpigenTek, EP-A63656). After three washes in PBS, samples were incubated with Alexa Fluor secondary antibodies with Hoechst (Sigma-Aldrich) for 45 min. Slides were mounted with Fluoromount-G Mounting Medium (Invitrogen). For PAX7 (Santa Cruz Biotechnology, sc-81648) and Ki-67 (Abcam, sp6 ab16667) staining, slides were incubated after fixation with 4% PFA in Acetone:Methanol (1:1) solution for 6 min at −20 °C, before blocking with 10% BSA. For IgG uptake, samples were incubated with an anti-rat secondary antibody for 1 h at room temperature. Pictures were acquired using an Axio Imager 2 or a LSM800 confocal (Zeiss).

Dystrophin (DYS1 and DYS2), β-Dystroglycan and TMEM65 protein intensity were quantified as mean intensity of the pixels using ImageJ and applying a set threshold on the intensity of the signals in each image. For consistency, all images were acquired under the same exposure settings and with the same background substraction. ImageJ was also used to manually quantify cardio-myocyte size on a set of images spanning the left ventricle wall stained with an anti-laminin antibody. TMEM65 area was assessed by using Automatic Particle counting in ImageJ after having set the same threshold range on black and white pictures. All values were displayed as average of measurements of at least three biological replicates.

## Transmission electronic microscopy and quantification

Muscle samples were fixed in a solution composed of 2% paraformaldehyde, 2% glutaraldehyde, in 0.1 M cacodylate buffer

pH 7.4 and 0.002% calcium chloride, overnight at 4 °C. After a wash in 0.1 M cacodylate buffer and postfixation in 1% osmium tetraoxide in 0.1 M cacodylate buffer, samples were incubated in 2% aqueous uranyl acetate at 4 °C overnight. After rinsing twice in water, samples were dehydrated in increasing concentrations of ethanol, with the final dehydration in 100% acetone (twice, 10 min each). Samples were infiltrated with 50% acetone 50% Epon overnight in a tight closed container. Finally, they were infiltrated twice with pure Epon (1 h each at room temperature) and then placed in molds with fresh Epon (EMBed 812, Electron Microscopy Sciences, Cat 14 120). Blocks were heated at 56 °C for 48 h. Semi-thin (0.5 μm thick) and ultrathin sections (70 nm thick) were cut with a Leica UC7 ultramicrotome (Leica, Leica Microsystemes). Semi-thin sections were stained with 1% toluidine blue in 1% borax and the region of interest was selected under the microscope. Ultrathin sections of the selected region were collected on copper grids and contrasted with Reynold's lead citrate. Ultrathin sections were observed using a Tecnai12 electron microscope (Thermo Fisher Scientific) operating at 120 kV. Pictures (4 K × 4 K pixels) were taken with an OneView camera (Gatan). Amplitude of intercalated disks (ICD) was calculated from 5 to 10 independent TEM images per each biological sample ($n = 3$), only where sarcomeres were longitudinally sectioned. ICD length was calculated using ImageJ, as the width of the fold region as indicated by the arrow in Fig. 6J.

## Western blot

Cryosections from frozen muscles were homogenized with a dounce homogenizer in a lysis buffer containing 50 mM Tris-HCl, pH 7.4, 100 mM NaCl, 0,5% NP40 and Halt Protease inhibitor cocktail (Pierce). Samples were then centrifuged for 5 min at $1500 \times g$ and denatured at room temperature for 30 min with Laemmli buffer. Protein concentration was determined by Bradford assay (Pierce). Proteins were separated by electrophoresis (Nu-PAGE 4–12% Bis-Tris gel; Life Technologies) and then transferred onto nitrocellulose membranes (GE Healthcare) and labeled with primary antibodies and secondary antibodies coupled to horseradish peroxidase. The primary antibodies used are anti-dystrophin (Leica, DYS1) and anti-nNOS (Thermofisher, Catalog #61-7000). Signals were visualized with SuperSignal West Pico Chemiluminescent substrate (Pierce). Images were acquired with Chemidoc MP (Biorad). Western blot quantifications were done by ImageJ calculating the pixel density of each band and expressing the quantified values as the ratio of band of interest to loading controls. Three independent biological samples were analyzed.

## snRNAseq

Cardiac apexes, including a portion of right and left ventricles and septum, were minced in 1 ml cold lysis buffer (1 mM Tris-HCl pH 7.5, 10 mM NaCl, 3 mM MgCl$_2$, and 0.1% Nonidet™ P40 in Nuclease-Free Water) and lysed for 3 min at 4 °C. The lysates were dounced with 10 strokes by adding 9 ml of cold wash buffer (PBS, BSA 2% and 0.2 U/μl RNase inhibitor from Roche) and filtered with 70 and 40 μm cell strainers. Nuclei were pelleted for 5 min at $500 \times g$ at 4 °C and washed with cold wash buffer, before staining them with DAPI (10 μg/ml). Nuclei were FACS sorted with a BD FACSAria III and around 20,000 nuclei were loaded into the 10x Chromium Chip. We followed the Single-Cell 3′ Reagent Kit v3.1 manufacturer protocol for GEM-Reverse Transcription, cDNA amplification and libraries preparation. Libraries were sequenced using an Illumina Nextseq 500 device. The bioinformatical analysis were performed using Cell Ranger pipelines, while clustering and differential gene expression evaluations were performed in Loupe. Cluster annotation was performed using expression of cell type specific markers (Fig. EV2). We obtained 9964 nuclei from WT, 6810 nuclei from BMD, and 7890 nuclei from DMD sample with an average of 940 genes per nucleus.

## RNA extraction and RT-qPCR

RNA extraction was carried out utilizing TRIzol™ Reagent and with mirVana™ miRNA Isolation Kit from Thermo Fisher Scientific and reverse transcription was performed using SuperScript III Reverse Transcriptase (Invitrogen). For quantitative PCR (qPCR), a StepOnePlus real-time PCR system from Applied Biosystems was utilized along with SYBR Green detection. To normalize the gene expression of the rat transcript, the expression levels were compared to the expression of *Hprt*. The primer sequences used for this analysis are as follows: *Hprt* For CAAGCTTGCTGCTG AAAAGGA, Rev TGAAGTACTCATTATAGTCAAGGGCATA TC; *Nppa* For TTTGGCTCCCAGGCCATATT, Rev TTCATCG GTCTGCTCGCTC; *Nppb* For TCCTTAATCTGTCGCCGCTG, Rev GGCGCTGTCTTGAGACCTAA; *Myh6* For CAAGAAGA ACTTGGTGCGGC, Rev TCATCGTGCATTTTCTGCTTGG; *Tmem65* For GGACTTGGCCTTGCAGGTTA, Rev CCAACAGC CTTGCCCAAATG; *Gja1* For CGCGATCCTTAACGCCTTTG, Rev CTCACGTCCCACGGAGAAAA; *Ryr2* For CCAACGCAGC AAGGAAAAGG; Rev CTGGCACTGAAGGTCTGGAG.

## Data correction and statistics

Data are presented as mean ± SD, and each dot on graphs represents a single biological replicate. The $p$ values are provided within the graphs only when a difference was statistically significant or close to $p = 0.05$. Each experiment was conducted on at least three independent biological samples. The Shapiro-Wilk test was used to test for normal distribution of data ($p < 0.05$). For data included in Figures, comparisons involving more than two groups with two variables were analyzed by a one-way ANOVA (except for Fig. 2B,C, two-way) with Fisher's LSD test, or Dunn's uncorrected test if the data distribution was not normal. Organ weights and force measurement values were corrected for interindividual variation in growth using the "tibia length" cube (TL$^3$) value, preferred to the body weight that is misleading in dystrophic conditions. The correcting formula was adapted from Hagdorn et al (2019). This method divided the considered variable by the sum of the TL$^3$ and a correction factor, calculated on the basis of the linear relationship between organ weight and TL$^3$. To present the data on a biologic scale, the corrected value was multiplied by the median of the TL$^3$. GraphPad Prism Software (versions 9.0 and 10.2) was used for statistical analyses and graph generation.

## Data availability

The snRNAseq data has been deposited in the Gene Expression Omnibus (GEO) database under the accession number GSE233158. R-BMDdel45-47 rats can be obtained through material transfer agreement by contacting the corresponding author at valentina.-taglietti@inserm.fr and frederic.relaix@inserm.fr.

The source data of this paper are collected in the following database record: biostudies:S-SCDT-10_1038-S44319-024-00249-9.

## Peer review information

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

## Acknowledgements

We thank the "Institut Clinique de la Souris"–PHENO MIN for the establishment of the BMD rat mutant line, the ImagoSeine core facility of Institut Jacques Monod for transmission electronic microscopy, Gilles Renault and Franck Lager (Plateforme Imageries du Vivant, Institut Cochin) for

echocardiographic data acquisition and analysis. We acknowledge financial support from the "Association Française contre les Myopathies" (AFM) via TRANSLAMUSCLE I and II programs (projects 19507 and 22946), the "Fondation pour la recherche médicale" (FRM, grants EQU20200301021 and SPF20170938733), the "Agence Nationale pour la Recherche" (ANR, grants SenoMuscle ANR-21-CE13-0006 and LabEx REVIVE ANR-10-LABX-73) and the ImagoSeine core facility (France-BioImaging and IBiSA).

## Author contributions

**Valentina Taglietti**: Conceptualization; Data curation; Formal analysis; Funding acquisition; Validation; Methodology; Writing—original draft; Writing—review and editing. **Kaouthar Kefi**: Data curation; Formal analysis; Methodology. **Busra Mirciloglu**: Methodology. **Sultan Bastu**: Methodology. **Jean-Daniel Masson**: Data curation. **Iwona Bronisz-Budzyńska**: Methodology. **Vassiliki Gouni**: Methodology. **Carlotta Ferri**: Methodology. **Alan Jorge**: Methodology. **Christel Gentil**: Methodology. **France Pietri-Rouxel**: Data curation; Methodology. **Edoardo Malfatti**: Data curation; Methodology. **Peggy Lafuste**: Conceptualization. **Laurent Tiret**: Conceptualization; Data curation; Formal analysis; Supervision; Validation; Investigation; Visualization; Methodology; Writing—original draft; Project administration; Writing—review and editing. **Frederic Relaix**: Conceptualization; Resources; Data curation; Supervision; Funding acquisition; Writing—original draft; Project administration; Writing—review and editing.

Source data underlying figure panels in this paper may have individual authorship assigned. Where available, figure panel/source data authorship is listed in the following database record: biostudies:S-SCDT-10_1038-S44319-024-00249-9.

## Disclosure and competing interests statement

The authors declare no competing interests.

# Expanded View Figures

**Figure EV1. Dystrophin immunofluorescence staining.**

(A) Localization of dystrophin epitopes recognized by the antibodies used in the study (NCL-DYS1 and NCL-DYS2). The BMD deletion of exons 45–47 at the level of the nNOS binding site is highlighted. ABD, Actin-binding domain; CR, Cysteine rich domain; CT, C-terminal domain; R, rod spectrin-like repeats. (B, C) Representative images of DYS1 (red), DYS2 (purple), and Laminin (cyan) immunostaining on TA (B) and heart (C) of WT, BMD, and DMD rats at 11 months. Scale bar 20 μm. (D, E) Quantification of dystrophin intensity detected by DYS1 (D) and DYS2 (E) antibodies in both TA and heart of WT, BMD, and DMD rats at 11 months. One-way ANOVA, $n = 3$, corresponding to the number of independent rats. Data are represented as mean ± SD.

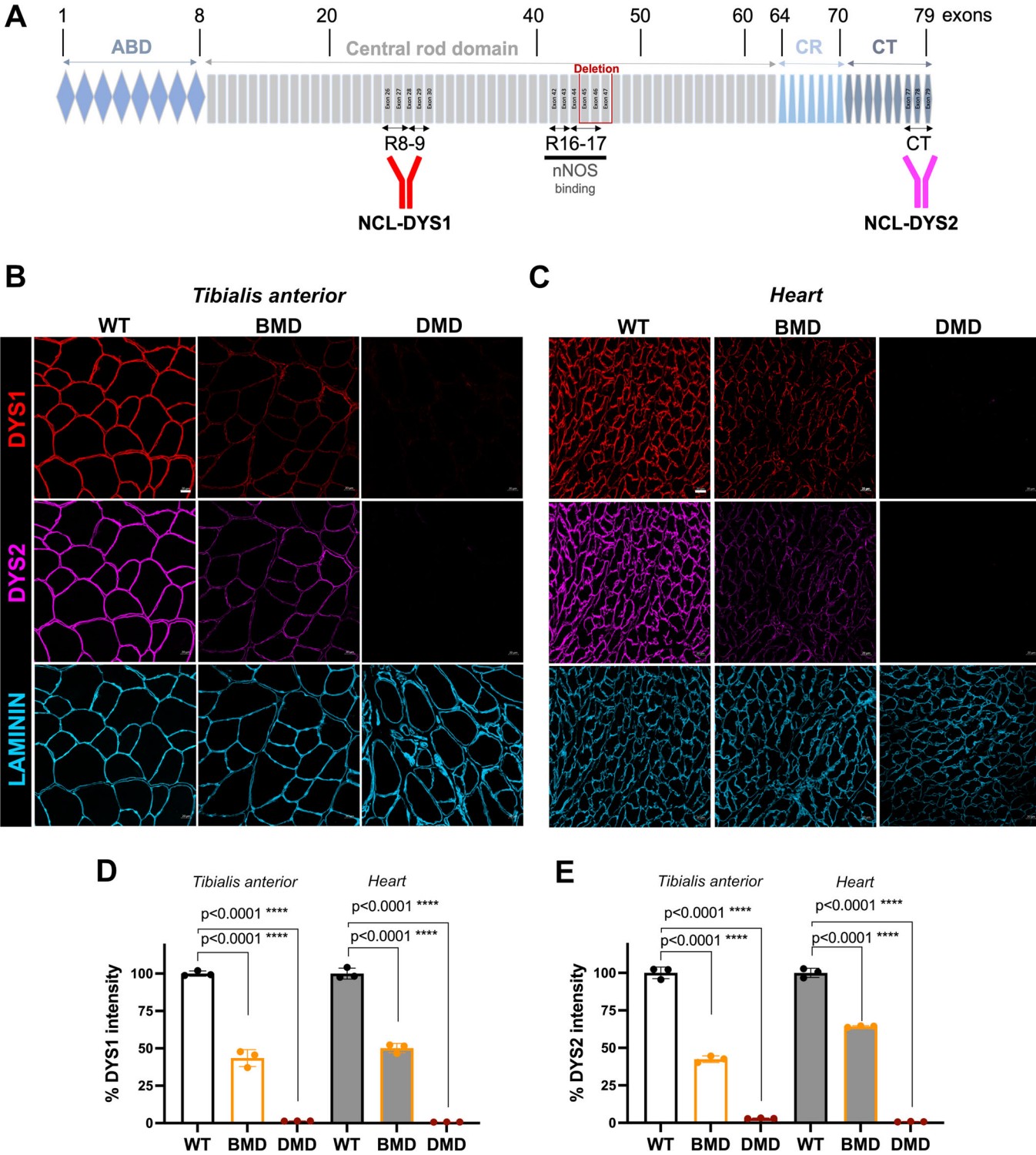

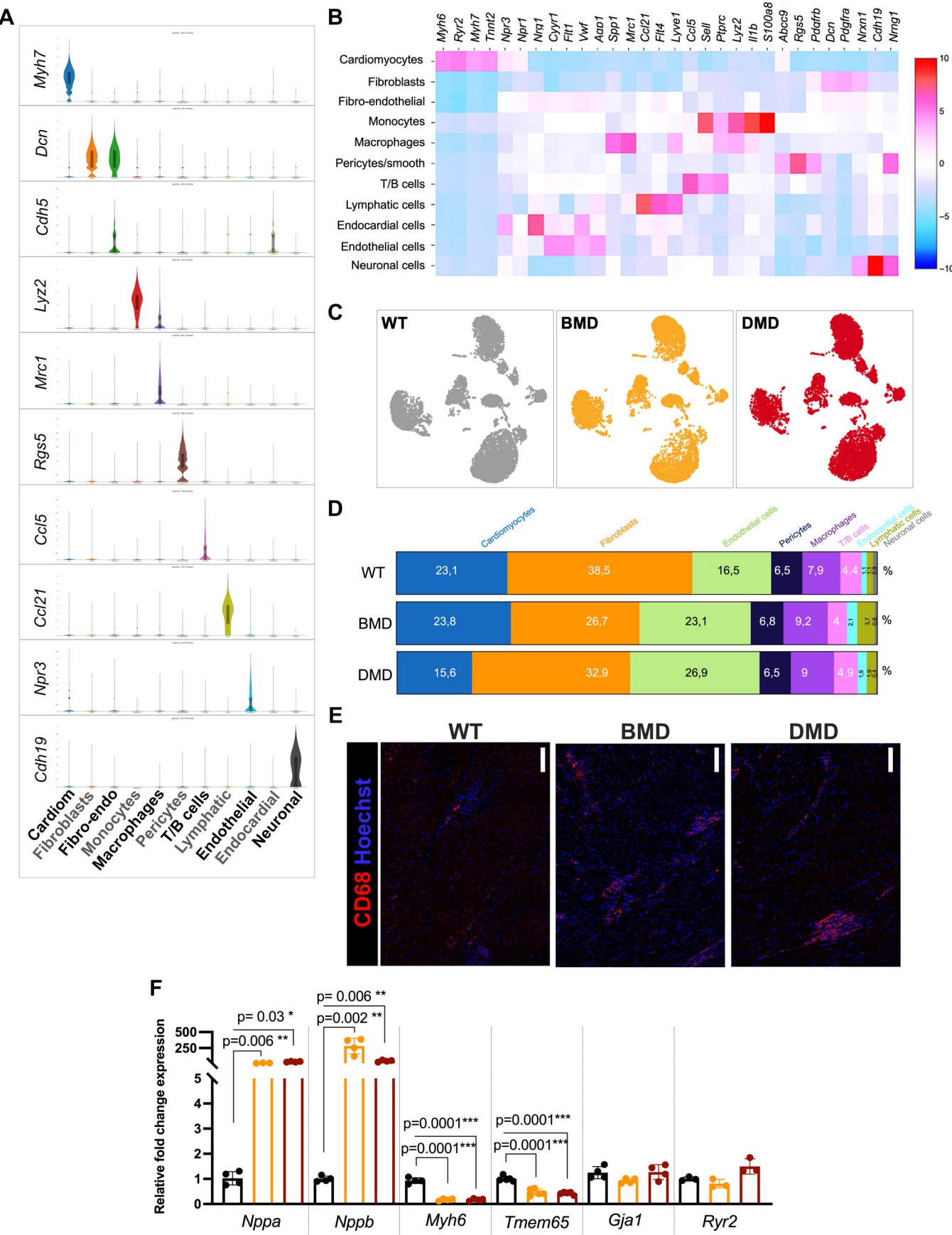

◀  **Figure EV2.  snRNAseq analysis and validations.**

(A) Violin plots showing the expression of Myh7 as marker of cardiomyocytes, Dcn for fibroblasts and fibro-endothelial cells, the latest marked also by the expression of Cdh5, Lyz2 for monocytes, Mrc1 for macrophages, Rgs5 for pericytes, Ccl5 for T and B cells, Ccl21 for lymphatic cells, Npr3 for endothelial cells and Cdh19 for neuronal cells. (B) Heat map of expression of differentially expressed genes identified from snRNA-seq. (C) UMAP representation of WT, BMD, and DMD snRNAseq datasets. (D) Proportions of cells (%) identified per sample. (E) CD68 immunostaining on left ventricles from WT, BMD, and DMD rats at 11 months of age. Scale bar 100 μm. (F) RT-qPCR analysis of Nppa, Nppb, Myh6, Tmem65, Gja1, and Ryr2 on ventricle tissue RNA extraction from 12-month-old WT, BMD, and DMD rats. One-way ANOVA, $n = 3$–4, corresponding to the number of independent rats. Data are represented as mean ± SD.

