## [Peer Review File · EMBO Reports]

Progressive cardiomyopathy with intercalated disc disorganization in a rat model of Becker dystrophy

Valentina Taglietti, Kaouthar Kefi, Busra Mirciloglu, Sultan Bastu, Jean-Daniel Masson, Iwona Bronisz-Budzyńska, Vassiliki Gouni, Carlotta Ferri, Alan Jorge, Christel Gentil, France Piétri-Rouxel, Edoardo Malfatti, Peggy Lafuste, Laurent Tiret, and Frédéric Relaix

Corresponding author(s): Valentina Taglietti (valentina.taglietti@inserm.fr) , Frédéric Relaix (frederic.relaix@inserm.fr)

Review Timeline:

Submission Date:	28th Nov 23
Editorial Decision:	6th Dec 23
Appeal Received:	19th Dec 23
Editorial Decision:	15th Feb 24
Revision Received:	10th Jun 24
Editorial Decision:	8th Aug 24
Revision Received:	9th Aug 24
Accepted:	22nd Aug 24

Editor: Deniz Senyilmaz Tiebe

Transaction Report:

Dear Prof. Relaix,

Thank you for submitting your manuscript to EMBO Reports. I have read your study carefully and discussed it in detail with the other members of our editorial team including our chief editor Dr. Bernd Pulverer, who mentioned discussing in depth with the first/dual corresponding author Dr. Valentina Taglietti in Crete. Furthermore, we sought advice from a good expert in the field whose opinion we trust. I regret to inform you that we have decided not to pursue publication of this manuscript in its current form, but we would be happy to send it out for full peer-review it with some additional analysis as mentioned below. In its current form, we recommend a transfer to our sister venue LSA, whose editors will send it out for external peer-review.

I apologize for this unusual delay in getting back to you, which was caused by the current high rate of new submissions to our office, affecting our usually much shorter editorial handling time. Also, we needed to wait for the expert's advice.

We appreciate your study developing a rat model of Becker muscular dystrophy and providing a functional and histopathological evaluation of these rats during their first year of life, in comparison to DMD rat models. The findings further reveal electrocardiographic abnormalities in BMD rats, which coincides with TMEM65 mislocalization in the ICDs. We realize that these findings are as such of interest to the field.

However, as mentioned above, I also sought advice from an external expert whose opinion we trust. In its current form, the advisor did not recommend further consideration of the manuscript at EMBO Reports. The advisor further noted:

'...I also think that reviewers are going to have an issue with the dystrophin staining and the fact that the authors don't even state the exons the antibody targets. This is a very addressable concern as there are entire panels of antibodies that should work targeting different sections of the dystrophin protein...'

As such, we concluded that the advance provided is not sufficient for publication in EMBO Reports in the current form of the manuscript. That said, we would be happy to reconsider the manuscript if you were to include such data, as this analysis will most likely be requested by the referees, in our view.

That said, and as mentioned above, your work in its current form is an excellent candidate for of our partner journal Life Science Alliance (<http://www.life-science-alliance.org/>; our broad scope Open Access journal published in partnership between the EMBO-, Rockefeller University-, and Cold Spring Harbor Laboratory Presses). The editors of Life Science Alliance would be pleased to send your manuscript for in-depth peer review; no reformatting is required. We very much hope you will be interested in this option: please follow the link below for transfer. Eric Sawey, Executive Editor of Life Science Alliance (e.sawey@life-science-alliance.org), will be pleased to answer any questions.

I very much hope that you are interested in this option - please use the following link for transfer; no reformatting is required.

Yours sincerely,

Deniz Senyilmaz Tiebe, PhD
Editor
EMBO Reports

** As a service to authors, EMBO Press provides authors with the ability to transfer a manuscript that one journal cannot offer to publish to another journal, without the author having to upload the manuscript data again. To transfer your manuscript to another EMBO Press journal using this service, please click on Link Not Available

Creteil, December 19th, 2023

Dear Dr. Deniz Senyilmaz Tiebe,

We are writing to resubmit our manuscript, EMBOR-2023-58557V1, entitled "TMEM65/connexin 43 cardiac alterations in a rat model of Becker muscular dystrophy," for reconsideration for publication in EMBO Reports.

In response to the concerns raised by an external expert, we have revised the manuscript and included an additional supplementary figure presenting novel immunofluorescence of Dystrophin with two distinct antibodies (DYS1 recognizing exons 26 to 30 and DYS2 targeting exons 77 to 79). The accompanying quantifications in both limb muscles and the heart demonstrate reduced dystrophin expression levels in our BMD rats, irrespective of the epitope targeted.

These new data, along with the revised manuscript, provide enhanced clarity and robustness to our findings on Dystrophin quantification, addressing the issues raised and reinforcing our observations.

We believe that these revisions significantly strengthen the manuscript. With these updates, we respectfully submit the revised version of our manuscript for your consideration.

We look forward to the possibility of seeing our manuscript published in EMBO Reports.

Best regards,

Prof. Frederic Relaix

PhD Valentina Taglietti

 Professeur Frédéric RELAIX
 Vice-Doyen recherche - UFR Santé
 Directeur Équipe RELAIX - Biology of the NeuroMuscular System
 INSERM U955 IMRB - Université Paris-Est Créteil
 8 rue du Général Sarrail - 94000 Créteil, France
 frederic.relaix@inserm.fr

Equipe/Team RELAIX
BIOLOGY OF THE
NEUROMUSCULAR SYSTEM

ESPRY research department

IMRB U955 INSERM/UPEC
 Faculté de santé
 8 rue du Général Sarrail
 94000 Créteil

DIRECTOR:
 Prof. Frédéric RELAIX

Vice-directors:
 Prof. F-Jérôme AUTHIER
 Dr. Philippos MOURIKIS

Group leaders
 Prof. Frédéric RELAIX
 Dr. Philippos MOURIKIS
 Prof. F-Jérôme AUTHIER
 Prof. Laurent TIRET
 Prof. Stéphane BLOT
 Prof. Hélène ROUARD
 Dr. Céline COLNOT

Contact
 Lila BENDAMECHE
 Gestionnaire d'équipe /
 Administrative support

Tél. +33 (0)1 48 98 46 03

Frédéric Relaix. Professor, UPEC - Paris Est-Créteil University - 94010 CRETEIL France
 Reference Center for Neuromuscular Disorders, Henri Mondor Hospital, AP-HP
 Vice-dean for research, UPEC Medical School, 8 rue du General Sarrail – 94000 CRETEIL, France
 Director, Team RELAIX – *BIOLOGY OF THE NEUROMUSCULAR SYSTEM*
 Head, Group 1 – Muscle stem cells, environment, development and preclinical modeling
Tel: (+33) 1 40 77 81 25 **e-mail:** frederic.relaix@inserm.fr

Dear Dr. Taglietti,

Thank you for the submission of your research manuscript to our journal, which was now seen by three referees, whose reports are copied below.

My apologies for this unusual delay in getting back to you. It took longer than anticipated to receive the full set of referee reports.

Referees express interest in the presented new rat model of Becker Muscular Dystrophy. However, they also raise significant concerns that need to be addressed to consider publication here.

Given these positive recommendations, we would like to invite you to submit a revised manuscript. Please revise your manuscript with the understanding that the referee concerns (as in their reports) must be fully addressed and their suggestions taken on board. Please address all referee concerns in a complete point-by-point response. Acceptance of the manuscript will depend on a positive outcome of a second round of review. It is EMBO reports policy to allow a single round of major experimental revision only and acceptance or rejection of the manuscript will therefore depend on the completeness of your responses included in the next, final version of the manuscript.

We realize that it is difficult to revise to a specific deadline. In the interest of protecting the conceptual advance provided by the work, we recommend a revision within 3 months. Please discuss the revision progress ahead of this time with me if you require more time to complete the revisions, or if you have questions or comments regarding the revision (also by video chat).

1. A data availability section providing access to data deposited in public databases is missing (where applicable).
2. Your manuscript contains statistics and error bars based on $n=2$. Please use scatter plots in these cases.

You can submit the revision either as a Scientific Report or as a Research Article. For Scientific Reports, the revised manuscript can contain up to 5 main figures and 5 Expanded View figures, and it should not exceed 27000 characters. If the revision leads to a manuscript with more than 5 main figures it will be published as a Research Article. In this case the Results and Discussion section should be separate. If a Scientific Report is submitted, these sections have to be combined. This will help to shorten the manuscript text by eliminating some redundancy that is inevitable when discussing the same experiments twice. In either case, all materials and methods should be included in the main manuscript file.

4) a .docx formatted letter INCLUDING the reviewers' reports and your detailed point-by-point responses to their comments. As part of the EMBO publication's Transparent Editorial Process, EMBO reports publishes online a Review Process File (RPF) to accompany accepted manuscripts. This File will be published in conjunction with your paper and will include the referee reports, your point-by-point response and all pertinent correspondence relating to the manuscript.

<https://www.embopress.org/page/journal/14693178/authorguide#transparentprocess>

5) a complete author checklist, which you can download from our author guidelines <https://www.embopress.org/page/journal/14693178/authorguide>. Please insert information in the checklist that is also reflected in the manuscript. The completed author checklist will also be part of the RPF.

6) Please note that all corresponding authors are required to supply an ORCID ID for their name upon submission of a revised manuscript (<<https://orcid.org/>>). Please find instructions on how to link your ORCID ID to your account in our manuscript tracking system in our Author guidelines <<https://www.embopress.org/page/journal/14693178/authorguide#authorshipguidelines>>

7) Before submitting your revision, primary datasets produced in this study need to be deposited in an appropriate public database (see <https://www.embopress.org/page/journal/14693178/authorguide#datadeposition>). Please remember to provide a reviewer password if the datasets are not yet public. The accession numbers and database should be listed in a formal "Data Availability" section placed after Materials & Method (see also <https://www.embopress.org/page/journal/14693178/authorguide#datadeposition>). Please note that the Data Availability Section is restricted to new primary data that are part of this study. * Note - All links should resolve to a page where the data can be accessed. *
If your study has not produced novel datasets, please mention this fact in the Data Availability Section.

Additional information on source data and instruction on how to label the files are available:
<https://www.embopress.org/page/journal/14693178/authorguide#sourcedata>

9) Our journal encourages inclusion of *data citations in the reference list* to directly cite datasets that were re-used and obtained from public databases. Data citations in the article text are distinct from normal bibliographical citations and should directly link to the database records from which the data can be accessed. In the main text, data citations are formatted as follows: "Data ref: Smith et al, 2001" or "Data ref: NCBI Sequence Read Archive PRJNA342805, 2017". In the Reference list, data citations must be labeled with "[DATASET]". A data reference must provide the database name, accession number/identifiers and a resolvable link to the landing page from which the data can be accessed at the end of the reference. Further instructions are available at <http://www.embopress.org/page/journal/14693178/authorguide#referencesformat>

- the name of the statistical test used to generate error bars and P values,
- the number (n) of independent experiments (please specify technical or biological replicates) underlying each data point,
- the nature of the bars and error bars (s.d., s.e.m.),
- If the data are obtained from n Program fragment delivered error `Can't locate object method "less" via package "than" (perhaps you forgot to load "than"?) at //ejpvfs23/sites23b/embor_www/letters/embor_decision_revise_and_review.txt line 56.' 2, use scatter blots showing the individual data points.

12) Please also note our reference format:

I look forward to seeing a revised version of your manuscript when it is ready. Please let me know if you have questions or comments regarding the revision.

Kind regards,

Deniz Senyilmaz Tiebe

Deniz Senyilmaz Tiebe, PhD
Scientific Editor
EMBO Reports

Referee #1:

This manuscript describes the generation and functional and pathological characterization of a new rat model of Becker muscular dystrophy (BMD). The significance of this model is that there are not really any useful models of the cardiomyopathy progression that causes most of the symptoms of BMD patients. Since BMD is quite rare, it is not feasible to conduct clinical trials for therapies that are not well tested in preclinical models of BMD, even though BMD represents a less severe form of Duchenne muscular dystrophy (DMD). In the characterization of this model, the authors find that TMEM65 and the gap junction protein connexin-43 show slightly altered localization that correlate with the presence of electrocardiographic abnormalities. Although this finding is highlighted in the title, it is really the development and thorough functional and pathological characterization of this new model of BMD that will have the most impact on the field. The comparison with the DMD rat model previously published by this group provides additional useful information for the muscular dystrophy field.

The experiments presented are thorough, well-controlled and complete. Figure are clearly laid out and well-labeled.

The only concern is the emphasis on connexin-43 in the title without any additional mechanistic insight. The minor abnormality of connexin-43 was shown in DMD cardiomyopathy mouse models almost 2 decades ago (15698839), albeit with much less sophisticated imaging at the time. This reviewer would suggest changing the title to something such as: Contrasting cardiomyopathy and muscle stem cell phenotypes between a new rat model of Becker muscular dystrophy and a Duchenne muscular dystrophy model. This title will likely lead to the published manuscript being more easily found by scientists looking for cardiomyopathy models or models of stem cell abnormalities.

There are a few minor typos or non-specific wording in the manuscript:

Page 2: variable extents - variable levels

Page 5, paragraph 2: DMD samples presented - DMD samples presenting

Page 5, paragraph 3: patients develops heart failure - develop

Page 6, paragraph 1: MD should be DMD

Page 8, paragraph 3: designed here as - designated here as

Referee #2:

Taglietti et al created a new rat model of Becker muscular dystrophy which they have characterized in this manuscript. This model would be helpful to the DMD/BMD research community as few models exist that can recapitulate the human phenotype. The main finding is that BMD rats have a more severe cardiomyopathy compared to DMD rats. This is significant as it bolsters the idea that DMD associated cardiomyopathy is not simply due to unhealthy skeletal muscle. However, this data is not entirely supported by the echocardiography data due to a low samples size.

Given that many comparisons are made between BMD and DMD in this manuscript, was statistical analysis performed between the BMD and DMD samples for all data? Reporting statistical analysis on BMD vs DMD data would be helpful - I only see this for a handful of graphs. In addition, since there is extensive data at two age groups, it would also be helpful to perform statistical analysis across the two ages to understand how disease progression affects the pathology.

Fig 1.

To make the case that there is more cytoplasmic signal in BMD and DMD rat muscles it would be helpful to see nNOS immunofluorescence data overlaid with laminin in Fig 1G. What are your thoughts on why there is higher mir34c and mir708 in BMD relative to DMD samples even though nNOS protein is similar or slightly elevated in BMD compared to DMD (Fig. 1I-J) and similar at the mRNA level (Fig. 1K)?

It would be helpful to see the immunofluorescence analysis data for DYS1 and DYS2 for cardiac tissue in S1 as shown for the TA. These would be the images used for quantification in S1C and S1D.

Fig 2.
The thickening of the ECM is apparent in the H&E sections however is less obvious in the Sirius red stained sections for BMD samples. Was sirius red stained tissue used to quantify the fibrotic area in Fig. 2D? Is there a reason why the quantification does not correlate with the data for Sirius red in Fig. 2A, especially if looking at the DMD samples at 12-mos relative to 6-mos? Based on the quantification of percent fibrotic area, it looks like the fibrosis in the BMD sections is similar at 6-mos and 12-mos of age and does not support the conclusion that BMD rats show greater fibrosis at 12-mos. The significant increase is only relative to WT samples which for the 12-mos groups shows less collagen deposition compared to WT at 6-mos. Since this is percent fibrotic area of each tissue, it is possible to perform a statistical analysis comparing across age of same genotype to make conclusions?

How was max force corrected in Fig 1I? The methods section does not describe this.

Fig 3.
Can you speculate on what are the Ki67 positive nuclei in panel A for BMD?
The conclusion that "no activated MuSCs were found in BMD samples" is not congruent with the quantification data in panel C at 12-mos of age for BMD samples where there is an outlier. Does this outlier represent a natural variability? Can you analyze additional samples?

Fig 4
For panel A and B, refer to comments to the fibrosis data in Fig.1.

The conclusion that there is a "significant increase in number of newly regenerated myofibers in BMD diaphragm at both 6 and 12 months" is not entirely supported by the data in D. Was statistical analysis performed between WT and BMD samples in Fig. 4D?

Fig. 5
Like in data for the TA and diaphragm, the percent area of fibrosis does not entirely align with the data in panel A.

Fig. 6
This analysis is looking for shared mechanism. What about the differences between the two models?
How to TMEM65 and CX43 co-localize?

Fig. 7
As was pointed out in the manuscript there is considerably variability. The sample size is too small given the variability of the echocardiography data. Some of the significant differences may be driven by the outlier data point(s), especially for the 12-mos BMD. These are challenging experiments to perform especially giving the sudden death and distress phenotypes in older animals. The data would be more conclusive if additional data point were included. A sample size of three is too few for BMD rats at 12-mos.

Tables:
Tables are missing proper legends.
What is the sample size used to generate the table?
Since organ weight is corrected with TL, it would be helpful to list the the TL

Referee #3:

The manuscript by Taglietti et al. describes the generation and initial characterization of a novel rat model for Becker muscular dystrophy (BMD), a milder form of muscular dystrophy, which although lethal is associated with mortality in patients in their 30s compared to late teenage age for patients suffering from Duchenne muscular dystrophy (DMD). Patients often develop severe heart pathologies such as cardiac arrhythmia and dilated cardiomyopathy (DCM) and often heart disease is causing death of BMD patients. A mouse model carrying the same deletion as the BMD rat reported in this paper develops mild muscular dystrophy and a DCM phenotype in ageing mice. The authors analysed first the skeletal muscle phenotype in limb muscle and diaphragm of the BMD rat displayed mild pathological aberrations at 12 months of age. However, in the BMD heart interstitial fibrosis, myocyte hypertrophy, and prolonged QT interval were found. Some transcriptional changes in both BMD and DMD hearts were observed indicating increased wall stress resulting in an elevation of Nppa, Nppb, downregulation of Myh6 and

upregulation of Myh7. The authors also observed a downregulation of TMEM65, which was not confirmed at the protein level. However, a disorganisation of the intercalated disk (ID) appears to be present in the BMD rat heart since both TMEM65 and Cx43 displayed a diffuse localisation at the ID. Finally, the authors found evidence for a dilation of the left ventricular chamber and decreased contractility.

The manuscript reports a very careful analysis of the striated muscle phenotype in this novel BMD rat model, which resembles very well the clinical phenotype of patients suffering from BMD. However, novel mechanistic insight into the pathogenic mechanism that is responsible for causing the DCM phenotype is missing.

1. While the authors decided to investigate the pathology at the molecular level with the help of snRNA sequencing of the ventricular apex, the results do not really provide much novel insight. Moreover, why was the apex used for this analysis and not the left ventricular free wall? Wall stress is most likely impacting the most in the free wall while the apex, might not experience this in the same way.
2. An analysis at the protein level would be far more informative as Dystrophin's effect on gene expression is secondary and will not provide insight into the underlying mechanism. However, early changes at the protein level might be very interesting and possibly provide novel interaction partners, which may specifically depend on the interaction with dystrophin domains deleted in the BMD rat model.
3. While identification of a distorted localisation of TMEM65 is interesting, it would be important to know if TMEM65 requires Dystrophin for targeting to the ID. Does Dystrophin and TMEM65 directly interact?
4. Moreover, it would be important to work out which compartments of the ID are affected by the BMD variant (PMID: 16406610) It would be important to perform transmission electron microscopy. The fact that N-cadherin appears unaffected suggests that only a specific subset of junctional complexes is probably affected by the BMD variant.
5. As you know the ID consists of many different ID proteins some of which are affecting each other. In this regard it would be interesting to look also into the sodium channel localisation at the ID since there is a mutual dependence of Cx43 and Nav1.5 for proper subcellular targeting at the ID.
6. Given the alterations at the ID in BMD rat hearts, it seems to be warranted to look at conduction velocity, which is the physiological parameter most likely altered in response to impaired Cx43 localisation.
7. Finally, it will also be important to find out about the localisation of Nav1.5 in the lateral myocyte membrane compartment. Previous work (PMID: 32536203) has shown that dystrophin plays an important role for sodium channel targeting specifically at the lateral membrane.

Responses to Reviewers

Referee #1:

This manuscript describes the generation and functional and pathological characterization of a new rat model of Becker muscular dystrophy (BMD). The significance of this model is that there are not really any useful models of the cardiomyopathy progression that causes most of the symptoms of BMD patients. Since BMD is quite rare, it is not feasible to conduct clinical trials for therapies that are not well tested in preclinical models of BMD, even though BMD represents a less severe form of Duchenne muscular dystrophy (DMD). In the characterization of this model, the authors find that TMEM65 and the gap junction protein connexin-43 show slightly altered localization that correlate with the presence of electrocardiographic abnormalities. Although this finding is highlighted in the title, it is really the development and thorough functional and pathological characterization of this new model of BMD that will have the most impact on the field. The comparison with the DMD rat model previously published by this group provides additional useful information for the muscular dystrophy field.

The experiments presented are thorough, well-controlled and complete. Figure are clearly laid out and well-labeled.

The only concern is the emphasis on connexin-43 in the title without any additional mechanistic insight. The minor abnormality of connexin-43 was shown in DMD cardiomyopathy mouse models almost 2 decades ago (15698839), albeit with much less sophisticated imaging at the time. This reviewer would suggest changing the title to something such as: Contrasting cardiomyopathy and muscle stem cell phenotypes between a new rat model of Becker muscular dystrophy and a Duchenne muscular dystrophy model. This title will likely lead to the published manuscript being more easily found by scientists looking for cardiomyopathy models or models of stem cell abnormalities.

There are a few minor typos or non-specific wording in the manuscript:

Page 2: variable extents - variable levels

Page 5, paragraph 2: DMD samples presented - DMD samples presenting

Page 5, paragraph 3: patients develops heart failure – develop

Page 6, paragraph 1: MD should be DMD

Page 8, paragraph 3: designed here as - designated here as

We thank the Reviewer for the time and effort in reviewing our work. We have carefully implemented the suggested changes, including adjusting the title to better suit the content and rectifying any identified typos.

The new title is: Progressive cardiomyopathy with intercalated disc disorganization in a rat model of Becker muscular dystrophy

Referee #2:

Taglietti et al created a new rat model of Becker muscular dystrophy which they have characterized in this manuscript. This model would be helpful to the DMD/BMD research community as few models exist that can recapitulate the human phenotype. The main finding is that BMD rats have a more severe cardiomyopathy compared to DMD rats. This is significant as it bolsters the idea that DMD associated cardiomyopathy is not simply due to unhealthy skeletal muscle. However, this data is not entirely supported by the echocardiography data due to a low samples size.

Given that many comparisons are made between BMD and DMD in this manuscript, was statistical analysis performed between the BMD and DMD samples for all data? Reporting statistical analysis on BMD vs DMD data would be helpful - I only see this for a handful of graphs. In addition, since there is extensive data at two age groups, it would also be helpful to perform statistical analysis across the two ages to understand how disease progression affects the pathology.

Fig 1.

To make the case that there is more cytoplasmic signal in BMD and DMD rat muscles it would be helpful to see nNOS immunofluorescence data overlaid with laminin in Fig 1G.

To provide a clearer picture of the subcellular distribution of nNOS and highlight differences between the groups, we updated Fig.1G with an overlay of nNOS immunofluorescence with laminin staining as requested.

What are your thoughts on why there is higher mir34c and mir708 in BMD relative to DMD samples even though nNOS protein is similar or slightly elevated in BMD compared to DMD (Fig. 1I-J) and similar at the mRNA level (Fig. 1K)?

The observed elevation of miR-34c and miR-708 in BMD despite similar nNOS protein levels compared to DMD suggests complex regulatory mechanisms beyond miRNA targeting. Multiple factors like translational regulation and protein stability may play a role in nNOS protein expression (Bender et al., 1999; Newton et al., 2003; Dunbar et al., 2004 ; Clapp et al., 2012). Additionally, the effectiveness of miRNAs might be influenced by context-dependent factors such as target site accessibility and compensatory mechanisms. Furthermore, miRNAs could exert broader effects by targeting upstream regulators or influencing protein stability pathways (Diener et al., 2024). Future studies investigating nNOS mRNA translation rates, protein degradation, and potential interactions with competing RNA-binding proteins or signaling pathways could provide a more comprehensive understanding of this discrepancy.

Bender AT, Silverstein AM, Demady DR et al (1999) Neuronal nitric-oxide synthase is regulated by the Hsp90-based chaperone system in vivo. *J Biol Chem* 274:1472–1478

Newton DC, Bevan SC, Choi S, Robb GB, Millar A, Wang Y, Marsden PA (2003) Translational regulation of human neuronal nitric-oxide synthase by an alternatively spliced 5'-untranslated region leader exon. *J Biol Chem.* Jan 3;278(1):636-44. doi: 10.1074/jbc.M209988200. Epub 2002 Oct 25. PMID: 12403769

Dunbar AY, Kamada Y, Jenkins GJ, Lowe ER, Billecke SS, Osawa Y (2004) Ubiquitination and degradation of neuronal nitric-oxide synthase in vitro: dimer stabilization protects the enzyme from proteolysis. *Mol Pharmacol* 66:964–969

Clapp KM, Peng HM, Jenkins GJ et al (2012) Ubiquitination of neuronal nitric-oxide synthase in the calmodulin-binding site triggers proteasomal degradation of the protein. *J Biol Chem* 287:42601–42610

Diener C, Keller A, Meese E (2024) The miRNA–target interactions: An underestimated intricacy, *Nucleic Acids Research*, Volume 52, Issue 4, 28

It would be helpful to see the immunofluorescence analysis data for DYS1 and DYS2 for cardiac tissue in S1 as shown for the TA. These would be the images used for quantification in S1C and S1D.

As suggested by the Reviewer, the pictures of DYS1 and DYS2 immunostaining of heart sections have been now included in the new Fig. S1C.

Fig 2.

The thickening of the ECM is apparent in the H&E sections however is less obvious in the Sirius red stained sections for BMD samples. Was sirius red stained tissue used to quantify the fibrotic area in Fig. 2D?

Yes, Sirius red positive area was quantified in Fig.2D.

Is there a reason why the quantification does not correlate with the data for Sirius red in Fig. 2A, especially if looking at the DMD samples at 12-mos relative to 6-mos?

The Reviewer raised a valid point regarding the potential discrepancy between the quantification data and Sirius Red staining in Figure 2A, particularly for DMD samples at 12 months. While our quantification method analyzes multiple muscle regions to robustly capture an average level of fibrosis across the entire tissue, excepting a single image perfectly reflecting this average is challenging. We selected new areas to take pictures that visually better reflect these numbers.

Based on the quantification of percent fibrotic area, it looks like the fibrosis in the BMD sections is similar at 6-mos and 12-mos of age and does not support the conclusion that BMD rats show greater fibrosis at 12-mos. The significant increase is only relative to WT samples which for the 12-mos groups shows less collagen deposition compared to WT at 6-mos. Since this is percent fibrotic area of each tissue, it is possible to perform a statistical analysis comparing across age of same genotype to make conclusions?

To increase the robustness of our analysis we increased the number of analyzed samples (12 months; N=6). The new data is now included in the Fig.2D, showing no significant differences between WT and BMD at 12 months of age. The pictures of Sirius Red and the manuscript were modified accordingly.

How was max force corrected in Fig 1I? The methods section does not describe this.

We apologize for the lack of information. We clarified in the paragraph "Data correction and statistics" our correction method, as following: *"Organ weights and force measurement values were corrected for inter-individual variation in growth using the "tibia length" cube (TL3) value, preferred to the body weight that is misleading in dystrophic conditions. The correcting formula was adapted from Hagdorn et al. (Hagdorn et al, 2019). This method divided the considered variable by the sum of the TL3 and a correction factor, calculated on the basis of the linear relationship between organ weight and TL3. To present the data on a biologic scale, the corrected value was multiplied by the median of the TL3"*.

Hagdorn, Q.A.J., Bossers, G.P.L., Koop, A.-M.C., Piek, A., Eijgenraam, T.R., Van Der Feen, D.E., Silljé, H.H.W., De Boer, R.A. & Berger, R.M.F. A novel method optimizing the normalization of cardiac parameters in small animal models: the importance of dimensional indexing. *American Journal of Physiology. Heart and Circulatory Physiology*. 2019. Vol. 316, n° 6, p. H1552-H1557. DOI:10.1152/ajpheart.00182.2019.

Fig 3.

Can you speculate on what are the Ki67 positive nuclei in panel A for BMD? The conclusion that "no activated MuSCs were found in BMD samples" is not congruent with the quantification data in panel C at 12-mos of age for BMD samples where there is an outlier. Does this outlier represent a natural variability? Can you analyze additional samples?

We thank the Reviewer for this observation. Since, Ki-67+ cells in BMD are mainly interstitial cells (data not shown), we can speculate that those cells are either FAPs or macrophages. To validate this hypothesis, staining with specific markers for these cell types would be necessary. This will be done in

future studies evaluating the modifications in mesenchymal and immune cell population in DMD and BMD rat models.

Regarding the outlier in the quantification data at 12 months of age for BMD samples, we performed additional analysis of three more BMD samples at 12 months and none of them had PAX7+ cells co-expressing Ki-67. The outlier in BMD samples might indicate variability within animals or potentially a rat with a more severe muscle phenotype and/or damage not linked with the disease. Notably, we observed heterogeneity in the disease progression among BMD rats, with some rats dying around 12 months of age while others survived up to 2 years. This variability was also reflected in echocardiography measurements, with some rats exhibiting severely compromised heart functions at 12 months compared to others. We believe that with a larger cohort, this "outlier" would represent the extreme value of a continuum. This hypothesis will be evaluated in the coming years by analyzing additional animals in future preclinical evaluations.

Fig 4

For panel A and B, refer to comments to the fibrosis data in Fig.1.

We increased the number of samples analyzed and we updated the pictures in Fig. 4A to include more representative pictures of the fibrotic deposition within the diaphragm.

The conclusion that there is a "significant increase in number of newly regenerated myofibers in BMD diaphragm at both 6 and 12 months" is not entirely supported by the data in D. Was statistical analysis performed between WT and BMD samples in Fig. 4D?

We thank the Reviewer, and we have corrected the sentence: "*Immunofluorescence analysis for eMHC showed a significant increased number of newly regenerated myofibers in BMD diaphragm at 6 months, with the DMD samples showing the greatest number of eMHC-positive fibers (Fig. 4C-D), in accordance with the intense regeneration in a more severe dystrophic context.*"

Regarding statistical analyses: we analyzed the 3 groups by a significant ANOVA ($p < 0.0001$), then compared the values two by two with a Fisher test. The p values shown on the graph are those obtained from the Fisher test.

Fig. 5

Like in data for the TA and diaphragm, the percent area of fibrosis does not entirely align with the data in panel A.

We modified Fig. 5A accordingly.

Fig. 6

This analysis is looking for shared mechanism. What about the differences between the two models?

We appreciate the Reviewer's comments regarding the limited identification of differentially expressed genes (DEGs) between BMD and DMD models.

Our initial analysis focused on identifying genes with the most significant expression changes between BMD and DMD. However, as shown in the heatmap below (Figure 1), we revealed no substantial number of DEGs between BMD and DMD. Indeed, DEGs are deregulated in the same direction (down or up) in both models *versus* WT samples, even at varying degrees of fold changes. For example, genes in group A and B are up-regulated in both BMD and DMD compared to WT samples; while genes in the group C are down-regulated in both BMD and DMD. There are several potential explanations for this observation. BMD and DMD might share a significant degree of underlying molecular mechanisms,

leading to limited differential gene expression despite phenotypic differences. Also, the sensitivity of our analysis to detect subtle differences could be limited by sample size or methodology (total number of sequenced genes). Indeed, snRNAseq typically has lower sequencing depth compared to bulk RNA-seq. Thus, we plan to employ additional approaches (such as bulk RNAseq), to gain a more comprehensive understanding of the molecular landscape underlying BMD and DMD, as follow-up of this first characterization focusing on shared pathogenic mechanisms.

Figure1: Heatmap of differentially expressed genes between WT, BMD and DMD cardiomyocytes.

How to TMEM65 and CX43 co-localize?

TMEM65 and Cx43 co-localize at the intercalated discs (ICDs) of cardiomyocytes, which are specialized cell junctions crucial for communication and synchronized depolarization of heart cells. TMEM65 acts as a scaffold protein, providing a physical platform for NaV1.5 and this interaction was necessary for the localization of NaV1.5 and Cx43 to the ICD (Teng et al., 2022). This allows Cx43 to form functional channels for electrical and chemical communication between cardiomyocytes. This has been now included in the manuscript.

Teng ACT, Gu L, Di Paola M, Lakin R, Williams ZJ, Au A, Chen W, Callaghan NI, Zadeh FH, Zhou YQ, Fatah M, Chatterjee D, Jourdan LJ, Liu J, Simmons CA, Kislinger T, Yip CM, Backx PH, Gourdie RG, Hamilton RM, Gramolini AO. Tmem65 is critical for the structure and function of the intercalated discs in mouse hearts. *Nat Commun.* 2022 Oct 18;13(1):6166. doi: 10.1038/s41467-022-33303-y. PMID: 36257954; PMCID: PMC9579145.

Fig. 7

As was pointed out in the manuscript there is considerably variability. The sample size is too small given the variability of the echocardiography data. Some of the significant differences may be driven by the

outlier data point(s), especially for the 12-mos BMD. These are challenging experiments to perform especially giving the sudden death and distress phenotypes in older animals. The data would be more conclusive if additional data point were included. A sample size of three is too few for BMD rats at 12-mos.

We appreciate the Reviewer's suggestion. Within the revision time provided by the editor, we have been able to increase the number of WT, BMD and DMD rats, but no more than the animals available at this mature age (12 months). In addition, the number of animals that could be analyzed at this age is limited by our ethical approval committee given the severe phenotype. Because we had to exclude few animals with a too low heart rate (< 310 bpm), possibly interfering with the analyzed data, the final set of echo data now includes 10 WT, 6 BMD, and 8 DMD subjects. We agree that it constitutes a more robust panel of animals; the updated values have been included in Figure 7 and do not modify our initial conclusions: we confirm the presence of a significant cardiac dilation in BMD rats, characterized by ventricular thinning and a reduced shortening fraction.

Tables:

Tables are missing proper legends.

We implemented them.

What is the sample size used to generate the table? Data in Table 1 was generated from 4 animals. This is now indicated in the Legend.

Since organ weight is corrected with TL, it would be helpful to list the the TL

We added to the table the TL values for each animal group, as required by the Reviewer.

Table 1. Morphometric and Muscle Weight Data

Parameter	6 months			12 months		
	WT	BMD	DMD	WT	BMD	DMD
Body weight (g)	628.3 ± 40.4	613.15 ± 17.5	518.7 ± 27.8*	796.7 ± 52.4	746.7 ± 52.4	531 ± 61.2***
Tibia length (mm)	50.41 ± 0.73	50.95 ± 0.32	49.50 ± 1.23	50.41 ± 0.09	50.34 ± 0.04	50.30 ± 0.29
Body Mass Index (g/cm ²)	0.78 ± 0.03	0.79 ± 0.04	0.62 ± 0.03***	0.98 ± 0.07	0.94 ± 0.04	0.67 ± 0.7 **
Body length (cm)	28.3 ± 0.4	27.8 ± 1.1	28.7 ± 1.1	28.5 ± 0.3	28.1 ± 0.2	28.1 ± 0.2
Tibialis anterior corrected (g)	1.03 ± 0.08	0.96 ± 0.07	0.88 ± 0.09\$	1.25 ± 0.18	1.15 ± 0.04	0.80 ± 0.15 **
Heart corrected (g)	1.98 ± 0.11	2.21 ± 0.13*	1.93 ± 0.15	1.77 ± 0.03	1.99 ± 0.09**	1.83 ± 0.09
EDL corrected (g)	0.28 ± 0.07	0.26 ± 0.01	0.25 ± 0.05	0.31 ± 0.08	0.33 ± 0.05	0.16 ± 0.04*
Soleus corrected (g)	0.28 ± 0.03	0.28 ± 0.04	0.23 ± 0.6*	0.32 ± 0.04	0.32 ± 0.04	0.22 ± 0.03 *

Results are expressed as mean ± SD; N≥4.

P values are calculated referred to WT by Tukey Post-Hoc Test; * p<0.05 **p<0.01 ***p<0.001; \$ p values are calculated referred to WT by Kruskal-Wallis Test (non-parametric).

Referee #3:

The manuscript by Taglietti et al. describes the generation and initial characterization of a novel rat model for Becker muscular dystrophy (BMD), a milder form of muscular dystrophy, which although lethal is associated with mortality in patients in their 30s compared to late teenage age for patients suffering from Duchenne muscular dystrophy (DMD). Patients often develop severe heart pathologies such as cardiac arrhythmia and dilated cardiomyopathy (DCM) and often heart disease is causing death of BMD patients. A mouse model carrying the same deletion as the BMD rat reported in this paper develops mild muscular dystrophy and a DCM phenotype in ageing mice. The authors analysed first the skeletal muscle phenotype in limb muscle and diaphragm of the BMD rat displayed mild pathological aberrations at 12 months of age. However, in the BMD heart interstitial fibrosis, myocyte hypertrophy, and prolonged QT interval were found. Some transcriptional changes in both BMD and DMD hearts were observed indicating increased wall stress resulting in an elevation of Nppa, Nppb, downregulation of Myh6 and upregulation of Myh7. The authors also observed a downregulation of TMEM65, which was not confirmed at the protein level. However, a disorganisation of the intercalated disk (ID) appears to be present in the BMD rat heart since both TMEM65 and Cx43 displayed a diffuse localisation at the ID. Finally, the authors found evidence for a dilation of the left ventricular chamber and decreased contractility.

The manuscript reports a very careful analysis of the striated muscle phenotype in this novel BMD rat model, which resembles very well the clinical phenotype of patients suffering from BMD. However, novel mechanistic insight into the pathogenic mechanism that is responsible for causing the DCM phenotype is missing.

1. While the authors decided to investigate the pathology at the molecular level with the help of snRNA sequencing of the ventricular apex, the results do not really provide much novel insight. Moreover, why was the apex used for this analysis and not the left ventricular free wall? Wall stress is most likely impacting the most in the free wall while the apex, might not experience this in the same way.

We opted for the apex as it provides a more reproducible area for analysis as it is consistently sectioned in the same manner across all samples. Additionally, this region offers a comprehensive representation of cardiac alterations, encompassing both ventricles and the septum. Furthermore, by focusing on the apex, we were able to utilize the left ventricles from the same rats for both immunofluorescence and transmission electron microscopy analyses, thereby reducing the number of animals used in this study and staying within the animal number limits authorized by our project ethical committee. This approach aligns with ethical considerations and the principles of the 3Rs, minimizing the need for additional animal subjects while maximizing the depth of information obtained from each specimen.

This non-targeted method was instrumental in providing novel clues on TMEM65 mislocalization in cardiomyocytes of both BMD and DMD rat models.

From our point of view, this new molecular player has shed light on the specific role of dystrophin in the organization of adhesion complexes at the membrane, altered in its absence in DMD, but also in the presence of a residual amount of the truncated isoform (BMD). In this revised version, these data have been expanded by convincing electron microscopy images, and we look forward to seeing this work continued by us or colleagues, via analyses that will enable to precisely describe the molecular cascade between dystrophin deficiency and the defect in the addressing of functional coupling proteins between cardiomyocytes. We have modified the discussion accordingly.

2. An analysis at the protein level would be far more informative as Dystrophin's effect on gene expression is secondary and will not provide insight into the underlying mechanism. However, early changes at the protein level might be very interesting and possibly provide novel interaction partners, which may specifically depend on the interaction with dystrophin domains deleted in the BMD rat model.

We acknowledge the potential value of a full proteomic, or a more directed DAPC-linked approach in elucidating the underlying mechanisms of dystrophin deficiency. While the argument highlights that

dystrophin's effect on gene expression might be a secondary consequence, directly analyzing protein interactions, particularly regarding dystrophin pull-down, presents certain challenges in the context of our study, including dystrophin large size and complexity. Also, pull-downs will not provide information regarding the direct dystrophin interactors, thus requiring proximity labeling approaches/FRET analysis on mature cardiomyocytes. However, culturing mature cardiomyocytes in sufficient quantities for proteomic and FRET analysis remains a significant challenge. To address these limitations and delve deeper into protein interactions in a future project, we are developing a human induced pluripotent stem cell (hiPSC) system for generating mature cardiomyocytes. This hiPSC-derived model will allow us to perform proteomic analysis in a more controlled and scalable manner, potentially using co-immunoprecipitation coupled with mass spectrometry on a human setting. Thus, acknowledging the relevance of a proteomic approach for future studies, it falls beyond the scope of this first paper.

3. While identification of a distorted localisation of TMEM65 is interesting, it would be important to know if TMEM65 requires Dystrophin for targeting to the ID. Does Dystrophin and TMEM65 directly interact?

Indeed, our data demonstrate that TMEM65 requires dystrophin for targeting to the ID. The DMD model (del52) is a knockout allele of the *Dmd* gene and in this condition, TMEM65 is mislocalized. We assumed you would like to know whether the interaction between dystrophin and TMEM65 is a direct molecular link via a specific interaction domain. Co-immunoprecipitation assays, proximity labeling approaches/FRET and a multistep validation procedure would be warranted to experimentally address this challenging question. To explore TMEM65 protein interactions, we conducted *in silico* predictions utilizing various interactome analysis systems, including STRING (Figure 2). From the available data used by the algorithm, there was no evidence of direct interactions between TMEM65 and dystrophin. This outcome was consistent across different interactome prediction methods, including whole interactome predictions and protein matching analyses between TMEM65 and proteins within the dystrophin complex. Although this computational prediction does not constitute a robust proof, it does point towards the more likely hypothesis of an indirect interaction between TMEM65 and dystrophin, or with target proteins of the dystrophin-associated complex that are activated to address proteins at the membrane.

Figure 2: STRING analysis (Version 10.5) showing the predicted molecular interactors of TMEM65.

4. Moreover, it would be important to work out which compartments of the ID are affected by the BMD

variant (PMID: 16406610) It would be important to perform transmission electron microscopy. The fact that N-cadherin appears unaffected suggests that only a specific subset of junctional complexes is probably affected by the BMD variant.

We sincerely appreciate the Reviewer's insightful suggestion to conduct transmission electron microscopy (TEM) analysis, which proved to be highly informative. The inclusion of TEM data in Figure 6J-K has provided valuable insights into the specific compartments of the intercalated discs (ICDs) affected in both BMD and DMD rats. Our findings reveal notable deterioration and disorganization of ICDs and adherent junctions, accompanied by disrupted interdigitation and detachment of sarcomeres from the junctions. These observations were consistent across both BMD and DMD samples. Furthermore, the wider amplitude of ICDs observed in BMD and DMD correlates with the more diffuse and blurred N-cadherin staining. The integration of these TEM findings significantly strengthens our data and enhances the quality of our paper. We are grateful for the Reviewer's constructive input, which has contributed to the robustness of our initial findings.

5. As you know the ID consists of many different ID proteins some of which are affecting each other. In this regard it would be interesting to look also into the sodium channel localisation at the ID since there is a mutual dependence of Cx43 and Nav1.5 for proper subcellular targeting at the ID.

We appreciate the Reviewer's insightful suggestion regarding the investigation of sodium channel localization at ICDs, particularly considering the mutual dependence between Cx43 and Nav1.5. We conducted staining with an antibody recognizing common epitopes of several sodium channels (pan antibody), which did not reveal any discernible alterations in sodium channel localization (Figure 3A) or expression (Figure 3B-C). We recognized the value of the targeted use of an antibody specific to Nav1.5, but the antibody validated in rats (SAB2107930) was out of stock and unavailable in the time frame of this revision.

However, as Nav1.5 is the main subunit of voltage-gated cardiac sodium channels, the absence of obvious alterations with Pan antibody suggests that the localization of this channel will not be profoundly altered in BMD and DMD rats, thus providing insight into your question.

Figure 3: A) Immunofluorescence of Pan-Na⁺ channel (purple) and Connexin 43 (Cx43, green) in left ventricles of WT, BMD and DMD rats at 12 months of age. Scale bar = 20µm. B) Western blot analysis of proteins extracted from WT, BMD and DMD left ventricles at 12 months of age for Pan-Na⁺ channel and Cx43. The analysis has been conducted on three biological replicates. Ponceau is used as loading control. C) Western blot quantifications of Pan-Na⁺ channel and Cx43 expression normalized on loading control.

6. Given the alterations at the ID in BMD rat hearts, it seems to be warranted to look at conduction velocity, which is the physiological parameter most likely altered in response to impaired Cx43 localisation.

We think we have investigated this aspect in Figure 5 of the manuscript through ECG analysis. At the basal level, ECG revealed no discernible alterations in conduction velocity in BMD rats compared to controls. However, when challenged with isoproterenol, the prolongation of the corrected QTpeak (QTpc) value, a feature of the ECG tracing of DMDDel52 rats even at rest, reflects a defect in electrical synchronization between cardiomyocytes, consistent with the asynchrony of wall contraction we were able to observe on echocardiography (Figure 7A). However, we did not observe any significant change in the PR segment, consistent with the absence of any major abnormality in the expression of fast sodium channels. Scariform junction defects are therefore subtle to the point of not altering overall depolarization wave conduction, but of producing synchronicity defects between cardiomyocytes, a hypothesis in line with reports of ventricular fibrillation in patients that we were able to observe in our rats.

7. Finally, it will also be important to find out about the localisation of Nav1.5 in the lateral myocyte membrane compartment. Previous work (PMID: 32536203) has shown that dystrophin plays an important role for sodium channel targeting specifically at the lateral membrane.

We appreciate the Reviewer's suggestion regarding Nav1.5 localization in the lateral myocyte membrane. As addressed in response 5, we conducted staining with a pan sodium channel antibody, which showed no discernible changes in localization (Figure 3A) or expression (Figure 3B-C).

Dear Valentina,

Thank you for submitting your revised manuscript. It has now been seen by two of the original referees. My apologies for this unusual delay in getting back to you, it took longer than anticipated to receive the referee reports.

As you can see, the referees find that the study is significantly improved during revision and recommend publication. However, I need you to address the points below before I can accept the manuscript.

- The figures should not be included in the manuscript text file and uploaded separately as one file per figure (both main figures and EV figures). The figure legends should stay in the text.
- Please provide 3-5 keywords for your study. These will be visible in the html version of the paper and on PubMed and will help increase the discoverability of your work.
- Please rename the Conflict of Interests section as "Disclosure Statement and Competing Interests".
- Please remove the Author Contributions section from the manuscript.
- Please make sure that the funding information in our manuscript tracking system is complete - we note that the following is not entered: the ImagoSeine core facility is a member of France-BioImaging (ANR-10-INBS-04) and IBSA, with the support of the "Who Am I" Labex, Inserm Plan Cancer, Region Ile-de-France and Fondation Bettencourt Schueller.
- We note the following regarding figure callouts: the two EV figures are not called out. Appendix figures (S1-S2) called out but there is no Appendix file.
- The manuscript sections should be in the following order: Title page - Abstract & Keywords - Introduction - Results - Discussion - Methods - Data Availability - Acknowledgments - Disclosure Statement & Competing Interests - References - Figure Legends - (Main Tables with legends) - Expanded View Figure Legends.
- Please resubmit source data as one zip file per figure.
- Please remove the reviewer access codes from the Data Availability section.
- Papers published in EMBO Reports include a 'synopsis' and 'bullet points' to further enhance discoverability. Both are displayed on the html version of the paper and are freely accessible to all readers. The synopsis includes a short standfirst summarizing the study in 1 or 2 sentences (max 35 words) that summarize the paper and are provided by the authors and streamlined by the handling editor. I would therefore ask you to include your synopsis blurb and 3-5 bullet points listing the key experimental findings.
- In addition, please provide an image for the synopsis. This image should provide a rapid overview of the question addressed in the study but still needs to be kept fairly modest since the image size cannot exceed 550 (width) x 300-600 (height) pixels.

Thank you again for giving us to consider your manuscript for EMBO Reports, I look forward to your minor revision.

Kind regards,

Deniz

--

Deniz Senyilmaz Tiebe, PhD
Senior Scientific Editor
EMBO Reports

Referee #2:

The others have satisfactory, addressed all issues. This manuscript will be important for the field to guide the use of more relevant animal models both for basic mechanistic research as well as translational studies. It is expected that the Rat model and cardiomyopathy will have a large impact on the field.

Referee #3:

The revised manuscript has greatly improved and the authors have addressed my suggestions. They performed some ultrastructural investigation of the intercalated disk region in WT, DMD, and BMD hearts. These interesting data confirmed the immunohistochemical investigations and provided further insights into the structural aberrations present in both DMD and BMD. Unfortunately, the attempts to perform NaV1.5 immunohistochemistry were not possible due to the unavailability of suitable antibodies. I appreciate the efforts of the authors to perform some preliminary data using a pan-specific sodium channel antibody, which did not reveal any discernible differences between WT, DMD, and BMD samples. Overall the manuscript has benefited from the additional data.

All editorial and formatting issues were resolved by the authors.

Valentina Taglietti
Univ Paris-Est Créteil, INSERM, U955 IMRB
8 rue du General Sarrail
Creteil F-94010
France

Dear Valentina,

Thank you for submitting your revised manuscript. I have now looked at everything and all is fine. Therefore, I am very pleased to accept your manuscript for publication in EMBO Reports.

Congratulations on a nice work!

Kind regards,

Deniz
--
Deniz Senyilmaz Tiebe, PhD
Senior Scientific Editor
EMBO Reports
